# Progressive unanchoring of Antarctic ice shelves since 1973

Bertie W. J. Miles[1 ✉] & Robert G. Bingham[1]

Mass loss of the Antarctic Ice Sheet has been driven primarily by the thinning of the floating ice shelves that fringe the ice sheet[1], reducing their buttressing potential and causing land ice to accelerate into the ocean[2]. Observations of ice-shelf thickness change by satellite altimetry stretch back only to 1992 (refs. 1,3–5) and previous information about thinning remains unquantified. However, extending the record of ice-shelf thickness change is possible by proxy, by measuring the change in area of the surface expression of pinning points—local bathymetric highs on which ice shelves are anchored[6]. Here we measure pinning-point change over three epochs spanning the periods 1973–1989, 1989–2000 and 2000–2022, and thus by proxy infer changes to ice-shelf thickness back to 1973–1989. We show that only small localized pockets of ice shelves were thinning between 1973 and 1989, located primarily in the Amundsen Sea Embayment and the Wilkes Land coastline. Ice-shelf thinning spreads rapidly into the 1990s and 2000s and is best characterized by the proportion of pinning points reducing in extent. Only 15% of pinning points reduced from 1973 to 1989, before increasing to 25% from 1989 to 2000 and 37% from 2000 to 2022. A continuation of this trend would further reduce the buttressing potential of ice shelves, enhancing ice discharge and accelerating the contribution of Antarctica to sea-level rise.

The contribution of the Antarctic Ice Sheet to global sea-level rise has been accelerating[7,8], enhancing the risk of flooding and other associated hazards to low-lying coastal communities[9]. Much of this mass loss has been attributed to warm ocean currents weakening the buttressing effects of its ice shelves[10–12], primarily in West Antarctica and the Wilkes Land coastline of East Antarctica[10–12], and hence driving an acceleration of ice discharge into the ocean[1,2]. Therefore, observational records that track the change in ice-shelf thickness over the longest possible time periods are important for explaining how the Antarctic Ice Sheet is changing and hence forecasting future mass loss. Existing records of ice-shelf thickness change derived from satellite altimetry span 30 years and have shown major ice-shelf thinning in some parts of West Antarctica and the western Antarctic Peninsula, thinning in the Wilkes Land sector of East Antarctica and limited change in most other ice shelves[3–5,12–14]. However, this satellite-altimetry record remains short relative to the typically multidecadal response times of many Antarctic ice shelves[15], and we do not know how widespread ice-shelf thinning was before 1992. Extending records of ice-shelf thickness change is important because a longer time series may directly help to lower uncertainties associated with the future contribution of Antarctica to global sea level by helping to calibrate numerical models[16,17].

Here, to extend the ice-shelf thickness change record to encompass the past 50 years, we implement a method that uses optical satellite imagery to track changes in the surface expression of pinning points that we treat as a proxy for ice-shelf thickness change (Fig. 1). Pinning points are common features around the Antarctic coastline that form when part of a floating ice shelf anchors onto a bathymetric high[6]: this interaction forms a bump on the otherwise smooth ice-shelf surface

that is visible in optical imagery. Crucially, for the analysis presented here, the surface expression of this bump changes through time as an ice-shelf thickens or thins[10,18] in response to its altering proportion of contact with the underlying bedrock high (Fig. 1). Although we focus here mainly on the monitoring of pinning-point change as a proxy for changes in ice-shelf thickness, we also note that pinning points are fundamentally important to ice-sheet mass balance because they buttress portions of upstream ice flow and limit ice discharge into the ocean[19–21]. The pinning points can also play an important part in calving by promoting rifting[22,23], and they influence the spatial pattern of basal melt by altering ocean circulation beneath ice shelves[24]. For all of these reasons, understanding better the evolution of pinning points through the longest possible time periods is important. Here we have systematically tracked changes in the surface expression of pinning points around Antarctica since 1973 to provide the first pan-ice-sheet observation-based characterization of Antarctic ice-shelf thickness change for the past five decades.

## Fifty years of pinning-point change

We used the full Landsat satellite-image archive to create two new near-cloud-free mosaics of the ice shelves of Antarctica for 1973 and 1989, with spatial resolutions of 60 m and 30 m, respectively (Methods and Extended Data Fig. 1). These two new mosaics represent our earliest near-cloud-free snapshots of the ice shelves of Antarctica. We used these mosaics, along with the existing Landsat-7 LIMA mosaic[25] from 2000, and Landsat-8 and Landsat-9 imagery from 2022, to track changes in the surface expression of pinning points from 1973–1989,

[1]School of GeoSciences, Edinburgh University, Edinburgh, UK. ✉e-mail: bertie.miles@ed.ac.uk

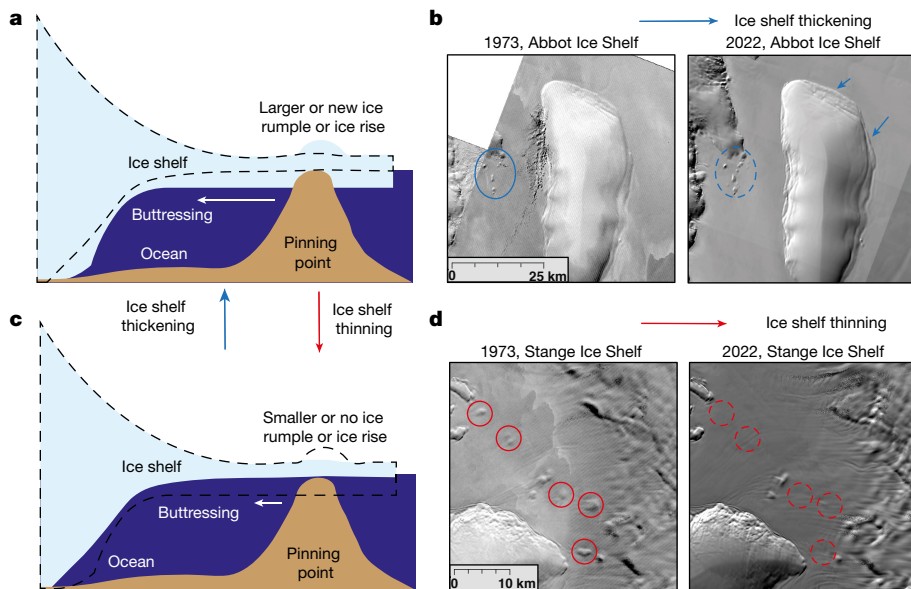

**Fig. 1 | Schematic of processes that cause changes in the surface expression of pinning points. a**, Ice-shelf thickening increases contact with the underlying bedrock high, causing the surface impression of the pinning point to increase in area. **b**, Example from Abbot Ice Shelf of ice-shelf thickening increasing the surface expression of pinning points between 1973 (Landsat-1 image) and 2022 (Landsat-8 image). **c**, Ice-shelf thinning reduces contact with the underlying bedrock high, causing the surface expression of the pinning point to decrease in area. The dotted lines represent the change in ice-shelf thickness. **d**, Example from Stange Ice Shelf of ice-shelf thinning reducing surface expression of pinning points between 1973 (Landsat-1 image) and 2022 (Landsat-8 image). Landsat imagery courtesy of the US Geological Survey. See Supplementary Animations for more examples. Scale bars, 25 km (**b**) and 10 km (**d**).

1989–2000 and 2000–2022, and to produce a long-term record of change between 1973 and 2022. We characterize the change in the surface expression of pinning points into three categories: smaller in extent, no detectable change and larger in extent.

A comparison between pinning-point change from 2000 to 2022 and ice-shelf thickness change derived from the ICESat and ICESat-2 satellites from 2003 to 2019 (ref. 13) shows a broad agreement in the spatial pattern of change (Methods and Extended Data Fig. 2). That is, the vast majority (86%) of pinning points that grew in area correspond to regions in which satellite altimetry recorded ice-shelf thickening (>0 m yr$^{-1}$), whereas 85% of pinning points that experienced no detectable change in area are in regions in which altimetry diagnosed limited change (between −1 m yr$^{-1}$ and 1 m yr$^{-1}$) in ice-shelf thickness. A smaller proportion (66%) of the pinning points that reduced in extent correspond to regions of altimetry-detected ice-shelf thinning (<0 m yr$^{-1}$; Extended Data Fig. 3). Several factors might have driven this reduced correlation, including the ungrounding of pinning points causing short-term localized thickening downstream in the wake of the former pinning points and our cautious approach to classifying pinning-point change, that is, classifying pinning points as no detectable change in which there is ambiguity (Methods). However, in some rare instances, we suggest that the visible retreat of the grounding line from Landsat imagery and extensive nearby pinning-point loss are incompatible with simultaneous ice-shelf thickening, for example, George VI Ice Shelf (Extended Data Fig. 4a). This suggests that in rare, highly localized examples, satellite altimetry may not be capturing the true direction in ice-shelf thickness change. Comparing two different altimetry products[3,13] in highly localized areas, we also observe conflicting signals in the direction of ice-shelf thickness change (Extended Data Fig. 4). Overall, however, the broad agreement between pinning-point change and satellite altimetry within the overlapping time frames, coupled with theoretical considerations (Fig. 1), substantiates the role of our pinning-point observations from 1973 to 1989 and 1989 to 2000 as a proxy for ascertaining the direction of ice-shelf thickness change.

Having demonstrated the validity of using changes to the area of pinning points surrounding Antarctica as a proxy for changes to ice-shelf thickness, we present, in Fig. 2, the first observationally constrained estimates of pinning-point change across Antarctica through the 1970s and 1980s, which, in turn, enable us to infer ice-shelf thickness changes around the ice sheet over the past 50 years. These observations demonstrate that ice-shelf thinning was generally less extensive around much of Antarctica than has been observed since the early 1990s onwards from satellite altimetry. However, the observations show that even between 1973 and 1989 concentrated hotspots of ice-shelf thinning were underway in Amundsen Sea Embayment in West Antarctica and in Holmes, Moscow University and Totten ice shelves in East Antarctica, demonstrating that these ice shelves began to thin at least 50 years ago. Pinning-point loss and ice-shelf thinning subsequently spread, and this is characterized by 15% of all mapped pinning points reducing in extent from 1973 to 1989, increasing to 25% in 1989–2000 and 37% between 2000 and 2022. In the following sections, we focus on the regional variations in pinning-point change across the Antarctic Peninsula, West Antarctica and East Antarctica.

In the Antarctic Peninsula, all pinning points were lost following the collapse of Prince Gustav, Larsen A, Larsen B and Wordie ice shelves over the past 50 years (ref. 26) (Fig. 3). Further south and facing the Weddell Sea, there has been very little change to pinning on Larsen C and Larsen D ice shelves (Fig. 3). The only exception has been Bawden Ice Rise, located at the front of Larsen C ice shelf. An ice rise is a type of pinning point that diverts ice flow around it and is characterized by maintaining its own local flow regime[6]. At this ice rise, basal melt rates have been increasing[14] and our observations flag visible shrinkage since 1989. The ongoing ungrounding from this ice rise will have an impact on local ice flow[27], but the absence of any major changes elsewhere on the Larsen C ice shelf would suggest that there is a limited prospect of catastrophic disintegration anytime soon.

Ice shelves facing the Bellingshausen Sea, incorporating those in both the western Antarctic Peninsula and in West Antarctica, exhibited contrasting changes in pinning points, which can be attributed to the varying drafts of these ice shelves. The relatively thin Wilkins[28] and Abbot ice shelves[29] thickened from 1973 to 1989 and 1989 to 2000 (Fig. 3d and

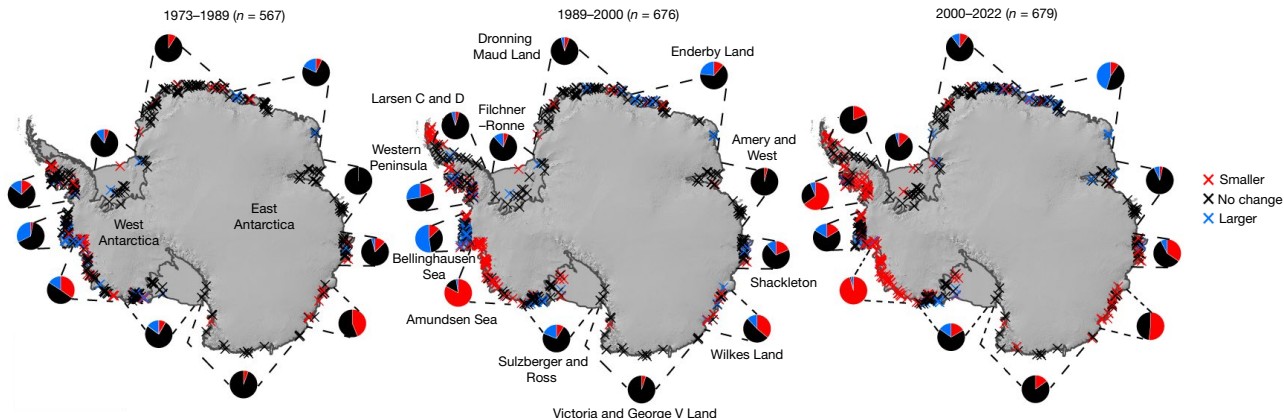

**Fig. 2 | Pinning-point change over three epochs spanning the periods 1973–1989, 1989–2000 and 2000–2022.** The pie charts represent the proportion of pinning points that reduced in area, remained the same or grew in area for each of the regions delineated by the dotted lines. The number of pinning points (*n*) mapped in each epoch is shown above each panel. Pie charts exclude data from collapsed ice shelves (Prince Gustav, Larsen A, Larsen B and Wordie). Mapped pinning-point change is overlain on the REMA mosaic of Antarctica[41].

Extended Data Fig. 5), whereas small sections of the relatively thick George VI[30], Stange and Venable[31] ice shelves were already thinning during this time period (Fig. 2). This implies that the thermocline on the continental shelf was deeper than the draft of Wilkins and Abbot ice shelves, but shallower than the drafts of George VI, Stange and Venable ice shelves allowing the warmer ocean water to facilitate basal melting. However, a clear shift in pattern occurred between 2000 and 2022, in which Wilkins and Abbott ice shelves transitioned to a more neutral pattern, with some pinning points continuing to grow, whereas others started to reduce in extent for the first time in the observational record. This is consistent with ocean observations from the 2000s, which place the thermocline at approximately the same depth as the mean draft of both Wilkins[28] and Abbot[29] ice shelves, which means the layer of warm water at the bottom of the ocean column would be able to intermittently reach the bases of these thinner ice shelves. At George VI, Stange and Venable ice shelves, much more widespread thinning took hold, with nearly every pinning point reducing in extent (Fig. 2). Collectively these patterns imply a decadal-scale raising of the thermocline depth and thickening of the layer of warm water on the continental shelf across the entire Bellingshausen Sea sector since 2000 that is consistent with ocean reanalysis products[32].

In the Amundsen Sea Sector in West Antarctica, 35% of pinning points reduced in area and 15% increased in area between 1973 and 1989 (Fig. 2). During this period, our results show that Pine Island Glacier, Thwaites, Dotson and Crosson ice shelves were already unanchoring from their pinning points and thinning decades before the earliest satellite-altimetry observations (Extended Data Figs. 6 and 7). This confirms that the processes driving the mass loss of West Antarctica have been underway for at least 50 years. Strong decadal variability in ocean forcing exists in this region and ocean-temperature data from the central tropical Pacific indicate that ocean conditions in the Amundsen Sea Sector were relatively cool during the mid-1970s to mid-1990s (ref. 11) (Extended Data Fig. 7), so we consider it that the pervasive thinning observed here was probably already well underway from before our records begin. This would be consistent with geological evidence that Pine Island Glacier began to retreat in the 1940s after its ice-shelf unanchored from a key pinning point[33], or possibly even earlier to coincide with the grounding-line retreat of Thwaites Glacier[34]. By contrast, further west on the Amundsen Sea coastline pinning points changed little between 1973 and 1989, with some even growing slightly in extent, and hence we did not detect evidence for ice-shelf thinning over much of the Getz Ice Shelf until the 1990s (Fig. 2 and Extended Data Fig. 7). Pinning-point loss markedly spread between 1989 and 2000,

with 83% of pinning points reducing in area, and only the far western section of Getz Ice Shelf escaping major pinning-point loss and ice-shelf thinning. In 2000–2022, 94% of all remaining pinning points in the Amundsen Sea Sector reduced in area (Fig. 2 and Extended Data Fig. 7), consistent with the widespread ice-shelf thinning diagnosed by satellite altimetry[3,13].

In Marie Byrd Land, amidst a general trend for little pinning-point change, our observations highlight notable pinning-point loss on Hull Glacier between 1973 and 1989. The presence of a heavily damaged ice tongue on Hull Glacier in 1973 may suggest a more prominent ice tongue in the years or decades before (Extended Data Fig. 6) and would imply that Hull Glacier was one of the few glaciers in Antarctica to be thinning in the 1970s. This long-term thinning may help to explain its recent rapid acceleration and grounding-line retreat[35]. At the heavily pinned Sulzberger Ice Shelf, very little change has occurred over the past five decades, although pinning-point loss near the grounding line (Fig. 2) suggests that warm water is now capable of reaching the grounding line and may represent the precursor to more widespread thinning across the ice shelf. Further west again, several pinning points grew substantially on the Swinburne Ice Shelf, in which we estimate that parts of the ice shelf have thickened upwards of 30 m (Extended Data Fig. 5). Across the Ross Ice Shelf, most pinning points changed very little, but Steershead Ice Rise and two other large ice rises located downstream of Kamb Ice Stream shrank consistently (Fig. 3e). Thus, our observations validate numerical models that have predicted a thinning of this section of the Ross Ice Shelf in response to the shutdown of Kamb Ice Stream[36]. We also observe between 2000 and 2022 a 5-km retreat of Engelhardt Ice Ridge, at the junction between Kamb and Whillans ice streams, continuing a longer-term retreat from at least the 1960s (ref. 37).

In East Antarctica, major pinning-point loss over the past 50 years has been concentrated around the fringes of Wilkes Land (Figs. 2 and 3). However, unlike for West Antarctica and the Antarctic Peninsula, there has been no clear acceleration in the proportion of pinning points reducing in area. There has been a reduction in pinning throughout each of our three epochs at Holmes Ice Shelf (Fig. 3c), highlighting the long-term thinning of this ice shelf. At Moscow University Ice Shelf, 6 km of erosion has occurred from an elongated ice rise that separates the ice shelf and the open ocean between 1973 and 2000. The shrinkage of this ice rise has been so extensive that it has allowed a new ice-shelf tributary to develop across its southern flank (Fig. 3b). Continued melting of this ridge could cause a major change in the flow direction of the entire ice shelf resulting in a substantial change in

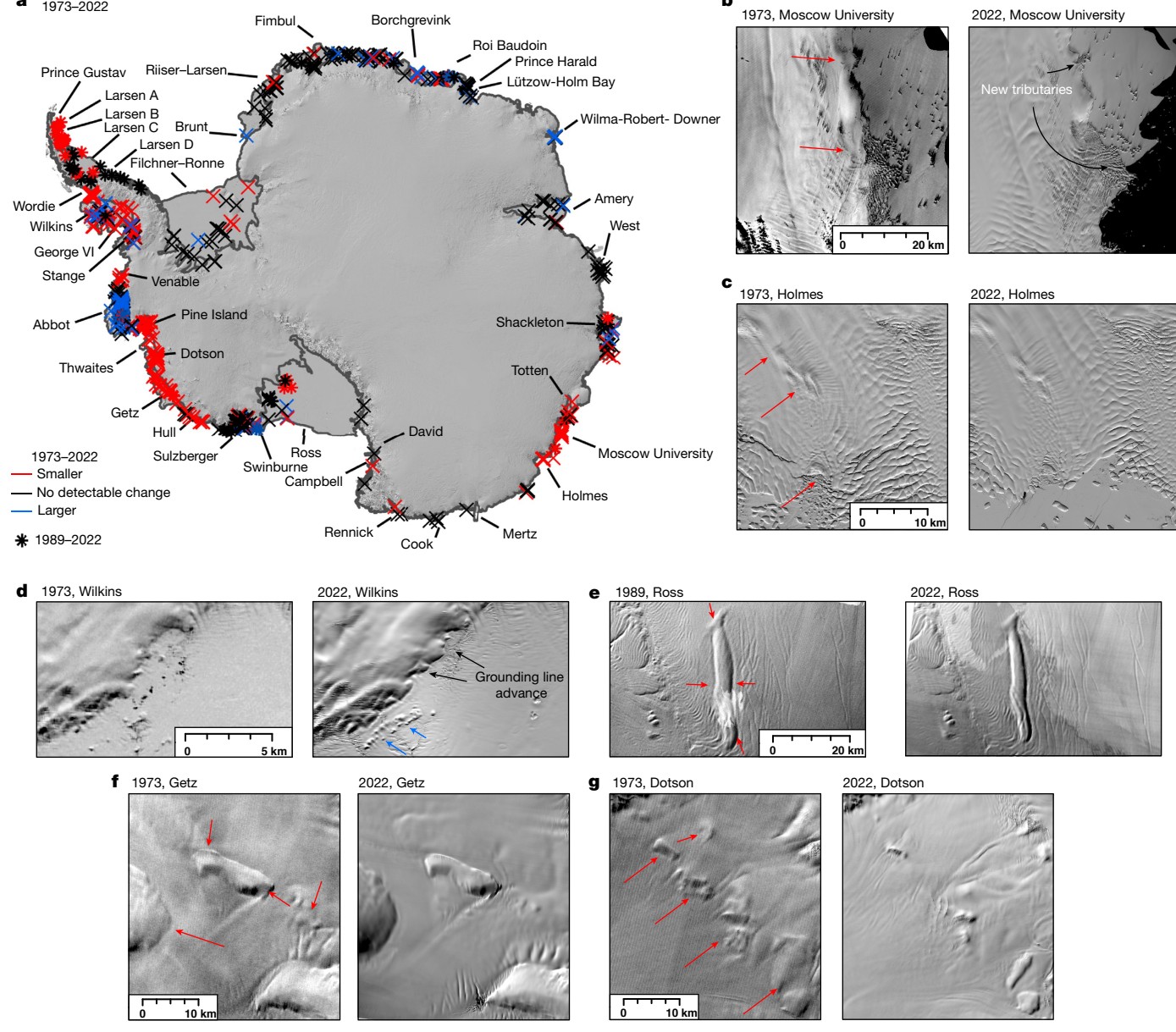

**Fig. 3 | Antarctic pinning-point change from 1973 to 2022. a**, Mapped pinning-point change from 1973 to 2022 overlain on the REMA mosaic of Antarctica[41]. Asterisks represent regions in which there was no cloud-free imagery available in 1973 and instead report on the change between 1989 and 2022. **b**–**g**, Landsat image pairs showing examples of pinning-point evolution.

The red arrows indicate pinning points that have reduced in area and the blue arrows indicate pinning points that have grown in area. Extensive examples of pinning-point mapping and animated images are located in the Supplementary Information. Landsat imagery courtesy of the US Geological Survey. Scale bars, 20 km (**b**,**e**), 10 km (**c**,**f**,**g**) and 5 km (**d**).

the dynamics of the entire Moscow University Glacier catchment. We observe more subtle losses of pinning at Totten Ice Shelf from 1973 to 1989 and 1989 to 2000, before more widespread pinning-point loss between 2000 and 2022 (Fig. 2). Outlet glaciers in Wilkes Land have been losing mass since the beginning of the satellite era[7]. Our results show that at least parts of their ice shelves were already thinning between 1973 and 1989. This hints that the initial trigger for mass loss and acceleration of outlet glaciers in Wilkes Land may have occurred pre-1973.

In Victoria Land and George V Land, we observe the loss of a major pinning point on the Campbell Glacier tongue and the shrinkage of a pinning point on the Rennick Ice Shelf in the 2000s, but limited change elsewhere (Fig. 2). The reduction in pinning of Rennick Ice Shelf is consistent with thinning in satellite altimetry[13] and, together with acceleration of the nearby Matusevich Glacier in the 2000s

(refs. 7,35), implies that warm water has recently reached this part of the Victoria Land coastline. At Shackleton Ice Shelf, most pinning points have not experienced substantive change, but we do observe some variability in a band of pinning points near its ice front. There has, however, been major unpinning on the nearby Conger Ice Shelf following its gradual retreat since 1973. In Enderby Land, there was limited change in pinning at the Wilma-Robert-Downer Embayment and in Lützow-Holm Bay in 1973–1989 and 1989–2000, but a growth of pinning points in these regions between 2000 and 2022 (Extended Data Fig. 5) confirms that these ice shelves have thickened. In Lützow-Holm Bay, thickening is consistent with the strengthening of easterly winds reducing the inflow of warm water underneath ice shelves[38]. Further west, some pinning-point loss was experienced at the front of Roi Baudouin Ice Shelf, and a pinning point disappeared from neighbouring Borchgrevink Ice Shelf. In Dronning Maud and

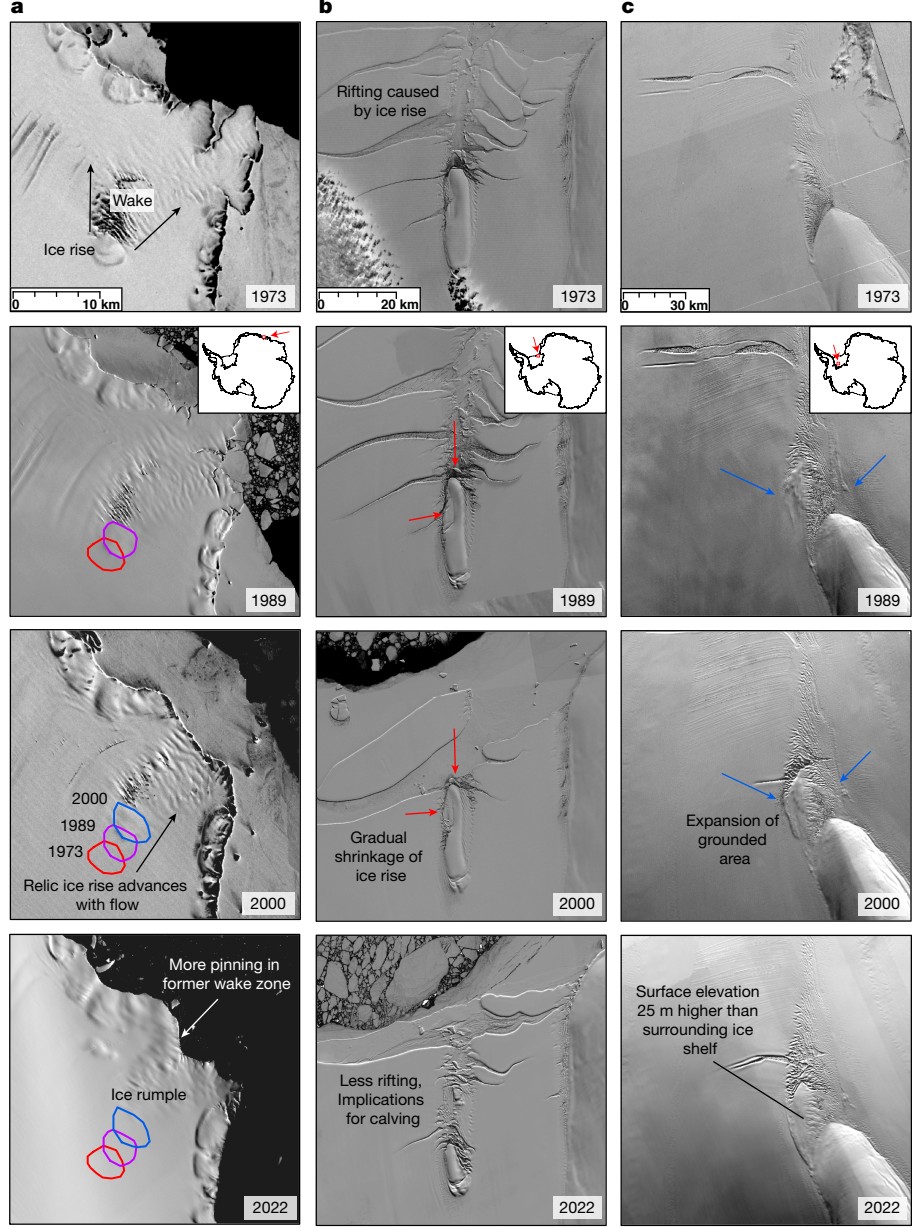

**Fig. 4 | Examples of ice-rise evolution from five decades of Landsat imagery.**
**a**, Borchgrevink Ice Rise. **b**, Hemmen Ice Rise. **c**, Korff Ice Rise. Red arrows indicate areas in which the ice rise has shrunk, and blue arrows indicate areas in which the ice rise has grown. Landsat imagery courtesy of the US Geological Survey. Scale bars, 10 km (**a**), 20 km (**b**) and 30 km (**c**).

Coats Land, we observed very few changes to most of the pinning points that fringe the coastline. Across Filchner–Ronne Ice Shelf, most pinning points also remained unchanged over the past five decades, although there were some notable changes to some of the prominent ice rises.

## Rapidly changing ice rises

The internal structures of ice rises have been crucial in reconstructing former ice-sheet flow and thickness change over centuries to millennia[39]. Some of our more striking observations are the particularly large breakups or growths of some ice rises, which provide insights into how these features evolve over decadal timescales. The 5-km-wide Borchgrevink Ice Rise ungrounded in the late 1970s (Fig. 4a), despite expressing limited ice-shelf thickness change in modern satellite-altimetry records[3,4]. This hints at vigorous ice-shelf thinning

occurring before its ungrounding in the late 1970s, strongly implying that the sub-ice-shelf bathymetry is conducive to warm-water intrusions[40]. After ungrounding, the 'relic' ice rise was transported downstream through the 1990s and 2000s before regrounding towards the ice front and forming the present-day ice rumple (Fig. 4a). In 1973, Hemmen Ice Rise, located at the front of Ronne Ice Shelf, was 22 km along its long axis, whereafter for the following three decades it gradually shrank before breaking apart in the mid-2000s (Fig. 4b). Before this break-up, it had played an important part in regulating the calving of Ronne Ice Shelf by promoting rifting[22,23]. Today at this location, there is now less rifting, meaning a change in calving behaviour might be expected over the coming decades. At Korff Ice Rise, also located on Ronne Ice Shelf, we observe a 20-km growth of the grounded section of the ice rise on its northern flank (Fig. 4c). Analysis of the Reference Elevation Model of Antarctica (REMA) Digital Elevation Model[41] shows that the surface of this newly grounded section is now around 25 m higher than the

surrounding floating ice shelf. This may represent the first stages of expansion of the entire Korff Ice Rise. Importantly, current numerical models do not account for any feedback associated with major changes in ice rises. Our observations show that these processes can happen relatively rapidly and on sections of ice shelves in which there has been limited thickness change over the satellite-altimetry era.

## Bleak future for some ice shelves

Our results have shown a marked, widespread and accelerating unanchoring of ice shelves from pinning points in the western Antarctic Peninsula and in the Amundsen Sea Sector over the past five decades (Fig. 3). Meanwhile, there has also been steady unanchoring of ice shelves from pinning points in the Wilkes Lands region of East Antarctica. The loss of many of these pinning points is likely to be permanent, owing to their hysteretic evolution[42], meaning that an ice-shelf thickening of a greater magnitude is required for pinning points to reform at comparable size. On multi-decadal timescales, this pinning-point loss may represent the first steps of irreversible ice-shelf loss and subsequent mass loss of the previously impounded ice sheet.

Our insight into the spatial pattern of ice-shelf thickness change in the 1970s and 1980s (Fig. 2) shows that ice-shelf thinning was already well underway in the Amundsen Sea Sector and Wilkes Land. After 1989, thinning has spread progressively across much of West Antarctica and the western Antarctic Peninsula, with previously unchanged pinning points reducing in extent through the 1990s to the present. We know that ice-shelf thinning is driven predominantly by warm modified Circumpolar Deep Water (MCDW) flooding the continental shelf and melting the bases of ice shelves[10,12]. However, the primary mechanism driving this progressive flooding of the continental shelf by warm MCDW over the past five decades remains unclear. There is some evidence that an anthropogenic-driven trend in winds over the continental-shelf edge in the Amundsen Sea may be driving an increase in MCDW transport onto the continental shelf since the 1920s (ref. 43). In East Antarctica, the poleward shift of MCDW since the 1930s in response to the poleward shift of westerly winds[44] may also be driving the thinning of ice shelves. Meanwhile, decadal-scale feedbacks emanating from the input of freshwater from ice-shelf melt onto the continental shelf may enhance the delivery of warm water beneath ice shelves on local scales[45], and the degree to which this may upscale to affect larger geographic scales continues to be investigated[46].

The overall acceleration of pinning-point loss is striking and paints a bleak future for many Antarctic ice shelves. From 2000 to 2022, the vast majority of pinning points in the 3,000-km stretch of coastline in West Antarctica from George VI Ice Shelf to Hull Glacier, along with an 800-km stretch of coastline in Wilkes Land, reduced in area or completely disappeared. Over the past 50 years, thinning at some of the most rapidly changing ice shelves means that they are close to being, or have already become, completely unanchored—for example, Thwaites Eastern Ice Shelf[47] and Pine Island Glacier[10]—which means that there is limited potential for further reductions in buttressing. Instead, the greatest concern may lie with those major ice shelves that are still substantially pinned but have shown clear signs of accelerated pinning-point loss. This includes George VI, Getz, Holmes, Moscow University and Totten ice shelves. A continuation of pinning-point loss in those locations will probably reduce buttressing and result in an acceleration in both ice discharge and mass loss.

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

## Methods

### Satellite imagery

We used the Landsat-1 and Landsat-2 satellite-image archive to create the earliest near-cloud-free mosaic of the ice shelves of Antarctica from the 1970s. Each image incorporated into the mosaic has a spatial resolution of 60 m. From a preliminary inspection of the imagery, Band 4 was determined to be the most suitable for pinning-point identification. There are relatively few repeat Landsat-1 and Landsat-2 image scenes and each footprint often had only a small handful of available images. Therefore, we downloaded all available images, including images with high cloud-cover percentages, because some still contained valuable cloud-free sections of ice shelves. The geolocation accuracy of the Landsat-1 and Landsat-2 imagery in Antarctica is poor and images were often offset by 10 s of kilometres. Therefore, we manually co-registered each image by tying stable features (for example, exposed bedrock or buried features in the grounded ice) to Landsat-8 and Landsat-9 imagery, which have excellent geolocation accuracy. For most Landsat-1 and Landsat-2 images, this gave co-registration accuracy to within 2 pixels. However, for a minority of images with few or no exposed bedrock, the co-registration accuracy may be greater than this. By filtering through all co-registered images and selecting the optimum cloud-free combination of imagery, we were able to produce a near-cloud-free mosaic covering nearly all ice shelves of Antarctica (Extended Data Fig. 2). The main exception to this is in the northern Antarctic Peninsula, in which there were no cloud-free images available over Larsen and Prince Gustav ice shelves. Elsewhere, the gaps were small and isolated. The mosaic consists of 251 images: 15% are from 1972, 59% from 1973, 19% from 1974, 6% from 1975, 1% from 1976 and less than 1% from 1978. For this paper, we ascribe to the mosaic a year stamp of 1973. A full list of the images used is available in Supplementary Table 1.

We undertook a similar process for the late 1980s ('1989') mosaic using Landsat-4 and Landsat-5 imagery. We used Band 2 to be consistent with the wavelength of Band 4 in the Landsat-1 and Landsat-2 satellites, and each image has a spatial resolution of 30 m. The geolocation accuracy of these images is also poor and required manual co-registration with the raw imagery offset by 100 s of metres. The optimum cloud-free combination of imagery provided complete coverage of all ice shelves apart from small isolated regions of cloud cover and the extreme southern sections of Filchner–Ronne and Ross ice shelves (Extended Data Fig. 1). The mosaic consists of 297 images: 8% are from 1986, 2% from 1987, 12% from 1988, 54% from 1989, 22% from 1990 and 2% from 1991. For the purposes of this paper, we ascribe a year stamp of 1989 to the mosaic. A full list of the images used is available in Supplementary Table 1.

For 2000, we used the Landsat Image Mosaic of Antarctica (LIMA)[25], which consists of cloud-free Landsat-7 imagery spanning 1999–2003 with a spatial resolution of 30 m. For 2022, we create a mosaic using Google Earth Engine[48]. We simply selected the most recent Landsat-8 or Landsat-9 imagery with cloud cover of less than 5%, with the earliest date of 1 January 2021. For the few regions in which cloud cover remained, we manually selected cloud-free Landsat-8 or Landsat-9 imagery to cover the small gaps. We used Band 3 to be consistent with earlier imagery that has a spatial resolution of 30 m.

### Pinning-point mapping

We primarily identified pinning-point locations using an existing inventory of ice rise and rumples[6,49]. Moreover, we used the MEaSUREs interferometry grounding-line product[50,51] to detect pinning points that were not included in reference[6]. We also manually identified several pinning points that were not included in either the ice rise and rumple dataset or the MEaSUREs product, and some pinning points that were present in the 1970s but have subsequently unpinned.

Each pinning point forms a bump on the otherwise smooth ice-shelf surface that is typically visible in optical imagery. The surface expression of this bump changes through time as an ice-shelf thickens or thins[10,18], in response to its altering proportion of contact with the underlying bedrock high (Fig. 1). For each of our epochs, we classified the change in the surface expression of each pinning point into three categories: growing in area, reducing in area or no detectable change. If a pinning point disappeared between two sets of images, we recorded it as reducing in area. We classified pinning-point change directly from each Landsat scene by magnifying each pinning point and finding the optimum contrast, before flicking between each successive epoch. For 1973 and 1989, there was often only one available cloud-free image over each ice shelf, making it impossible to be consistent in terms of choosing Landsat images with similar solar azimuth angles. In a small number of cases, extremes in lighting caused by differing solar zenith angles between mosaics may have had an impact on our classification of pinning-point change. In the small number of cases in which it is unclear if the surface expression of pinning points has changed, or if it simply reflects a change in atmospheric conditions (for example, shadows) or an artefact of comparing imagery of different qualities, we erred on the side of caution and classified them as showing no detectable change. However, for the vast majority of pinning points, the direction of change is obvious between the periods over which we examined change, and we include detailed examples of our classifications from every major ice shelf in Supplementary Fig. 1 and through animated imagery. Even for those images that were offset by greater than two Landsat-1 pixels (> 120 m) the change in shape of pinning points is clear.

### Comparison with satellite altimetry

We compared our pinning-point classification results with satellite-altimetry-based observations of ice-shelf thickness change over the overlapping time period. The longest available time series of satellite-altimetry-derived ice-shelf thickness change stretches from 1992 to 2018 (ref. 3), but this dataset is relatively coarse, with a spatial resolution of 10 km, and its coverage of some of the smaller ice shelves is limited. Instead we focus our comparison on the ice-shelf thickness change dataset derived from the ICESat and ICESat-2 satellites between 2003 and 2019 (ref. 13). This dataset has a higher spatial resolution (5 km), covers all floating ice shelves, and its timespan is a very close match to our 2000–2022 epoch of pinning-point change (Extended Data Fig. 2). We extracted the median thickness change from a 7.5-km buffer surrounding each pinning point in which there is available data (n = 467). The median thickness change is compared with the mapped direction of change for each pinning point. Overall 86% of pinning points that grew in size correspond to regions of ice-shelf thickening (>0 m yr⁻¹), 66% of pinning points that reduced in extent correspond to regions of ice-shelf thinning (<0 m yr⁻¹) and 85% of the pinning points that did not change notably in area correspond to regions of limited change in ice-shelf thickness classified as between −1 m yr⁻¹ and 1 m yr⁻¹ (Extended Data Fig. 3). The main reason why we do not detect change in some pinning points while the surrounding floating ice thins is related to the underlying topography beneath pinning points. A pinning point resting on flat topography would require relatively small reductions in ice-shelf thickness for ice to unground from a large surface area, resulting in large visual changes at the surface from optical imagery. This is in contrast to a pinning point with steep sides, in which relatively small reductions in ice-shelf thickness may cause only a very small area of the pinning point to unground, resulting in very small changes to the pinning-point area at the surface, that are not detectable in Landsat imagery.

### Data availability

All Landsat data used in this study are freely available to download from USGS Earth Explorer, and we include a list of all scenes used in the Supplementary Information. The co-registered Landsat scenes used

in the 1973 and 1989 mosaics[52] are available to download from https://doi.org/10.7488/ds/3810. The 2022 Landsat mosaic[53] is available to download from https://doi.org/10.7488/ds/7531. The shapefiles that map the direction of pinning-point change are available in the Supplementary Information and at https://doi.org/10.7488/ds/7583 (ref. 54). The inventory of ice rises and rumples[43] used to locate pinning points is available at https://doi.org/10.21334/npolar.2015.9174e644. The MEaSUREs Antarctic Grounding Line from Differential Satellite Radar Interferometry[44] is available at https://doi.org/10.5067/IKBWW4RYHF1Q. All mapping figures were produced using ArcMap 10.8.

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

**Acknowledgements** Landsat imagery was provided free of charge by the US Geological Survey Earth Resources Observation Science Center. B.W.J.M. was supported by a Leverhulme Early Career Fellowship (ECF-2021-484). R.G.B. acknowledges funding from the UK Natural Environment Research Council (NE/S006613/1). We thank A. Jenkins for providing the normalized ocean temperature index for the Amundsen Sea.

**Author contributions** B.W.J.M. and R.G.B. conceived the study. B.W.J.M. processed the satellite imagery, generated the 1973 and 1989 mosaics and mapped the pinning-point changes. Both authors analysed and interpreted the results. B.W.J.M. produced all the figures and wrote the paper, with contributions from R.G.B.

**Competing interests** The authors declare no competing interests.

**Additional information**
**Correspondence and requests for materials** should be addressed to Bertie W. J. Miles.

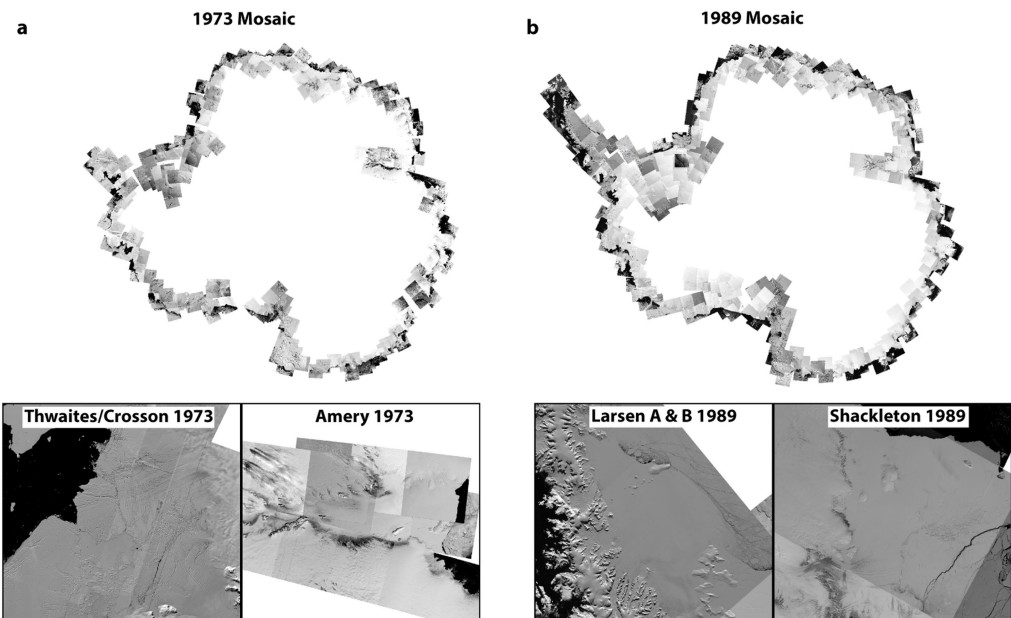

**Extended Data Fig. 1 | Mosaics of Antarctic ice shelves. a)** 1973 Landsat-1/Landsat-2 mosaic with example imagery from Thwaites Glacier/Crosson Ice Shelf and Amery Ice Shelf. **b)** 1989 Landsat-4 / Landsat-5 mosaic with example imagery from Larsen A and B Ice Shelves, along with Shackleton Ice Shelf. Landsat imagery courtesy of the U.S. Geological Survey.

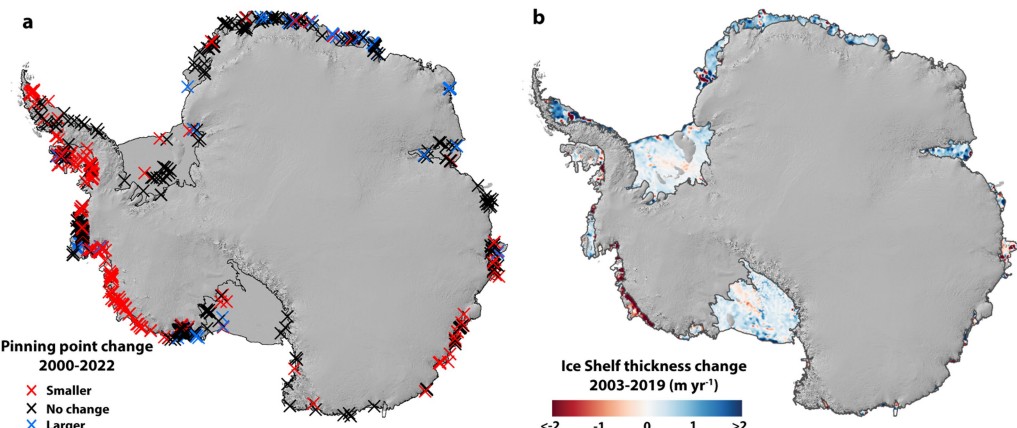

**Extended Data Fig. 2 | Comparison between mapped pinning point change and altimetry derived ice shelf thickness change. a)** mapped pinning-point change from 2000-2022 (this study) and **b)** ICESat derived ice-shelf thickness change from 2003-2019[13]. Note the broad agreement in the spatial pattern of change. In both panels data is overlain on the REMA mosaic of Antarctica[41].

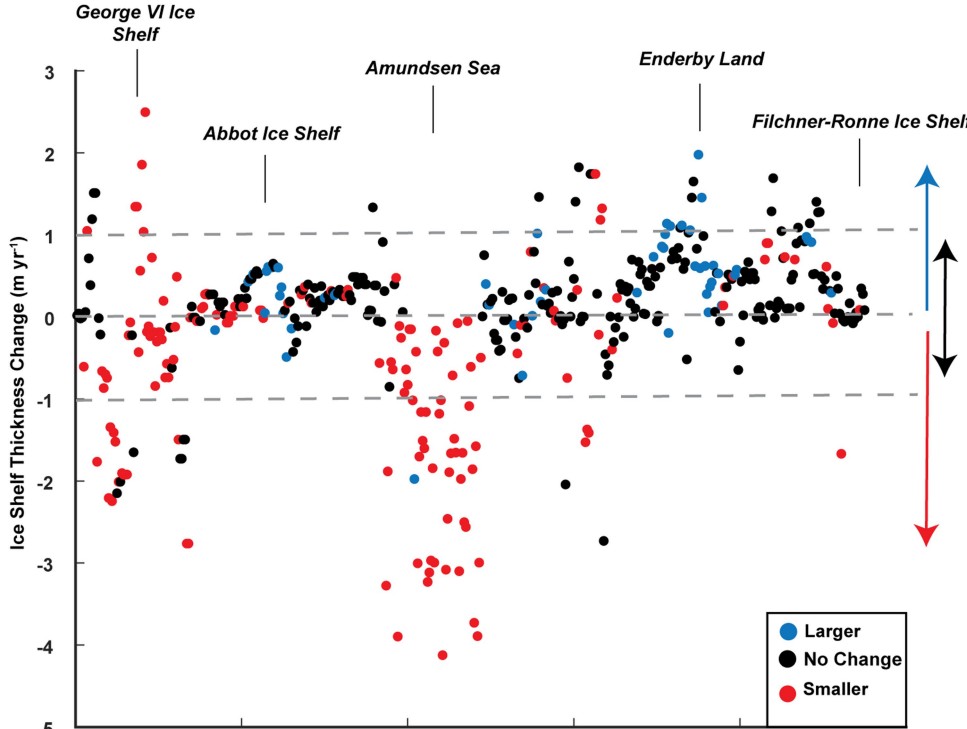

**Extended Data Fig. 3 | Comparison between ice-shelf thickness change and pinning points.** Satellite altimetry derived ice-shelf thickness change extracted from the vicinity of pinning points 2003–2019[13], with each data point colour coded in relation to mapped pinning point change 2000–2022. Data points are plotted from east to west, starting at the Antarctic Peninsula. The blue arrow on the right represents the quadrant for thickening (>0 m yr$^{-1}$), where 86% of growing pinning points are located (blue dots). The black arrows on the right represent the quadrant for limited thickness change (between −1 and 1 m yr$^{-1}$), where 85% of the pinning points that are not changing in size (black dots) are located. The red arrow on the right represents the quadrant for ice shelf thinning (<0 m yr$^{-1}$), where 66% of the pinning points that are reducing in size (red dots) are located.

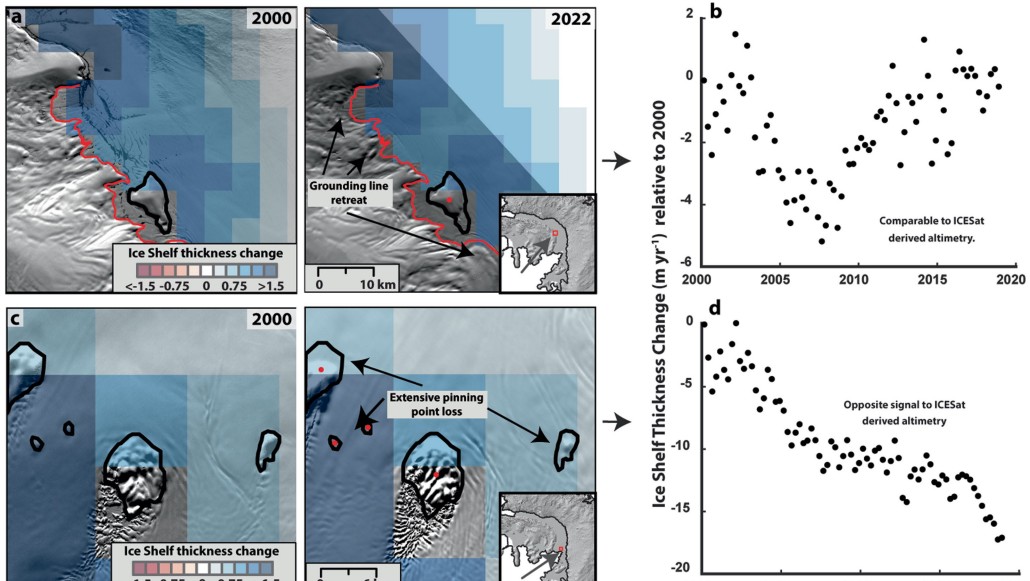

**Extended Data Fig. 4 | Examples of mismatches between pinning point mapping and ice shelf thickness change from satellite altimetry. a)** Ice-shelf thickness change between 2003 and 2019[13] from ICESat laser altimetry overlain on pinning-point change between 2000 and 2022 for Martin Ice Rise, George VI Ice Shelf. **b)** Time series of radar altimetry derived ice-shelf thickness change[3] averaged over the area shown in a and b. Satellite-altimetry products show thickening or limited change, but grounding-line retreat and extensive pinning-point loss identified here from the Landsat imagery suggest localised thinning. **c)** Same as a and b, but located over a cluster of pinning points at the southern ice front of George VI Ice Shelf. **d)** Time series of radar altimetry derived ice-shelf thickness change[3] averaged over area shown in d and e. Satellite-altimetry products differ over this area in terms of direction of ice-shelf thickness change.

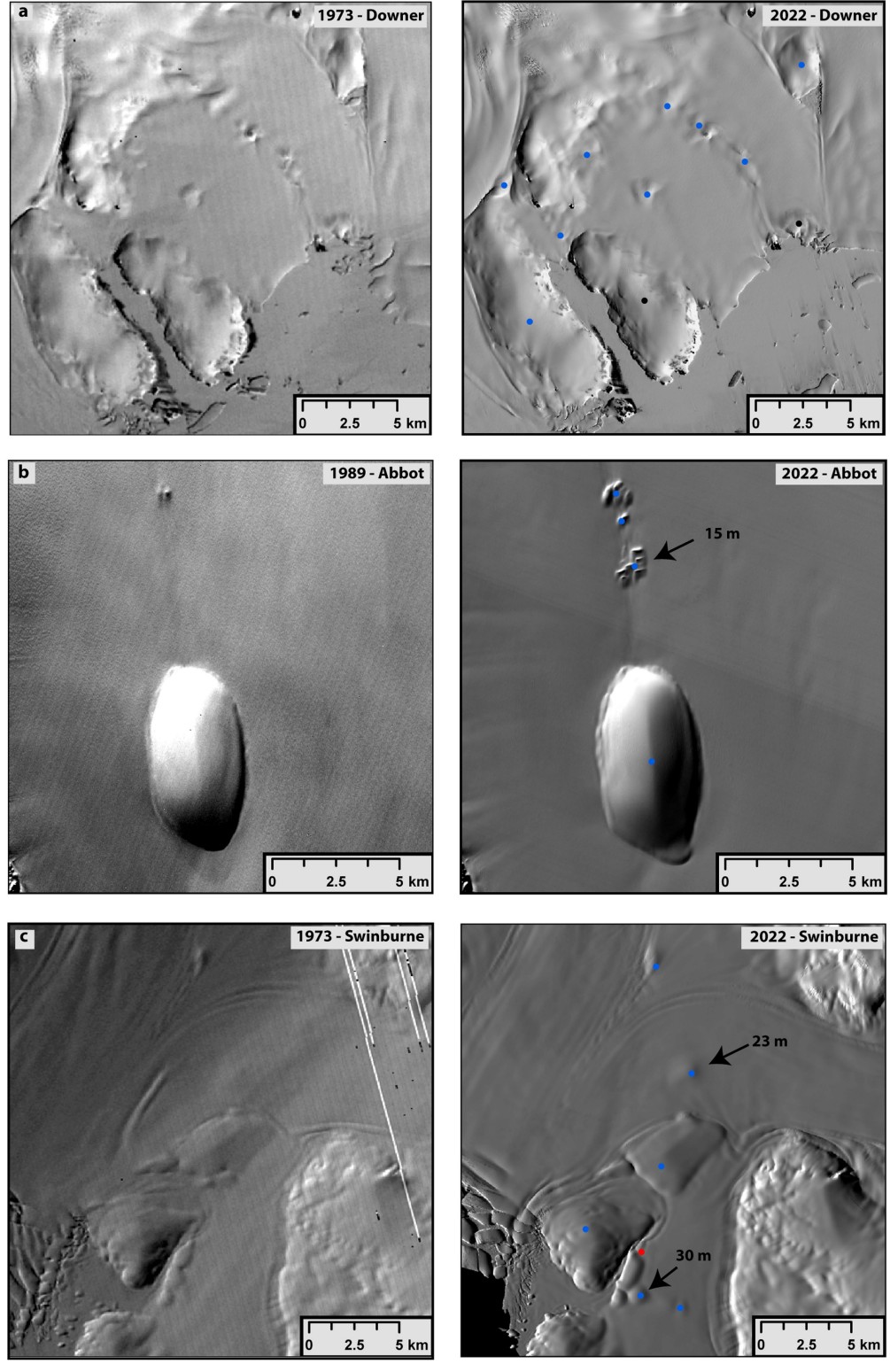

**Extended Data Fig. 5 | Examples of pinning-point growth from pairs of Landsat satellite images. a)** Wilma-Robert-Downer Glacier system, **b)** Abbot Ice Shelf and **c)** Swinburne Ice Shelf, with pinning-point change mapping overlain as small circles (coloured blue for growth, red for shrinkage). The numbers represent the difference in surface elevation between the pinning point and the surrounding flat ice shelf derived from the REMA DEM[41]. At these locations, this represents the minimum amount of ice-shelf thickening. Extensive examples of pinning-point mapping and animated images are located in the Supplementary Information. Landsat imagery courtesy of the U.S. Geological Survey.

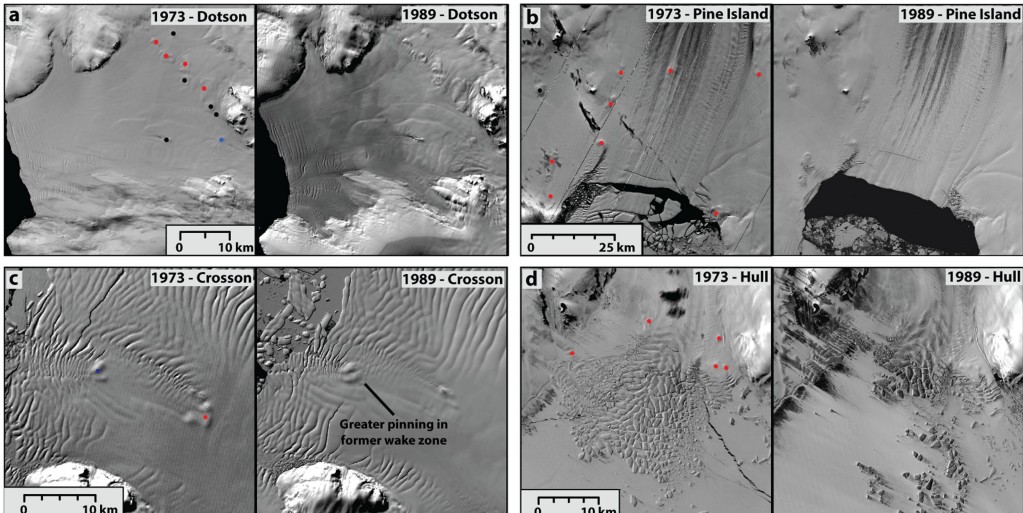

**Extended Data Fig. 6 | Examples of pinning-point change in West Antarctica between 1973 and 1989. a)** Dotson Ice Shelf, **b)** Pine Island Glacier Ice Shelf, **c)** Crosson Ice Shelf, **d)** Hull Glacier. Small circles mark pinning points mapped for this study (coloured red for shrinkage, blue for growth). Landsat imagery courtesy of the U.S. Geological Survey.

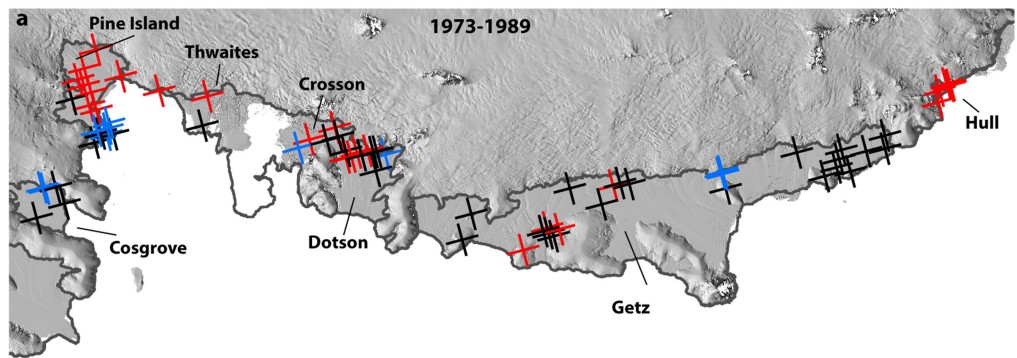

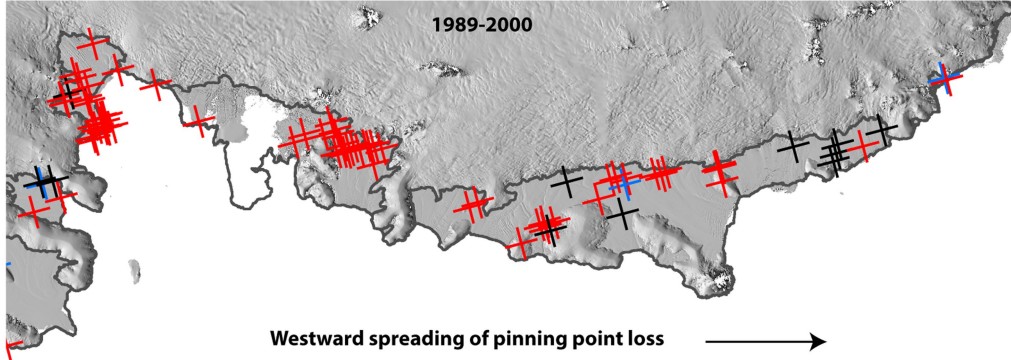

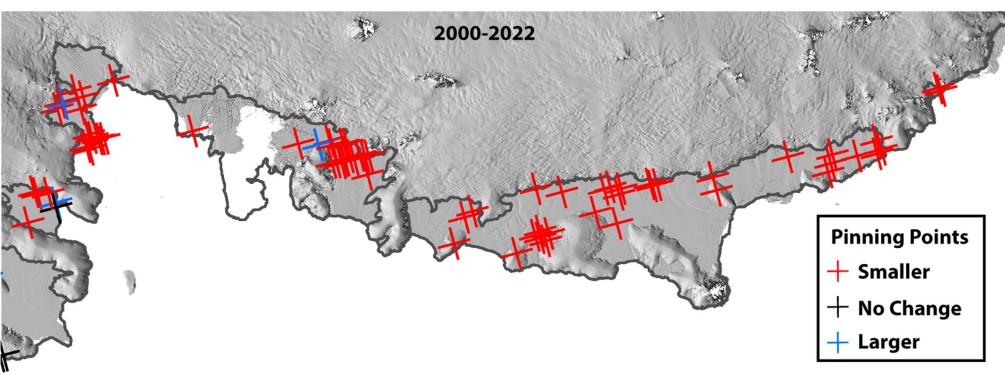

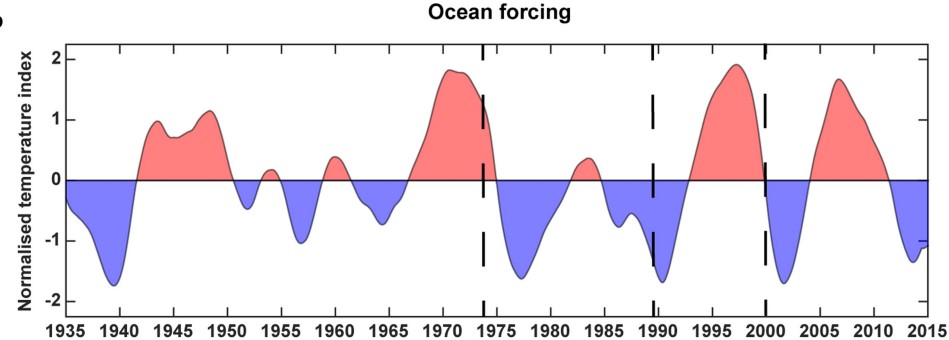

**Extended Data Fig. 7 | Pinning-point change in the Amundsen Sea Sector over the past five decades and ocean forcing reconstructions. a)** Mapped pinning point from 1973–1989, 1989–2000 and 2000–2022 overlain on the REMA mosaic[41]. Pinning-point loss spread westward over the past five decades.

**b)** Normalized ocean-temperature index for the eastern Amundsen Sea inferred from central tropical Pacific sea-surface temperatures[11]. Dotted lines represent the epoch boundaries in this study. The period 1973–1989 is characterised by relatively cool ocean forcing.