## [Peer Review File · Nature]

Manuscript Title: Progressive unanchoring of Antarctic ice shelves over multiple decades

Reviewer Comments & Author Rebuttals

Reviewer Reports on the Initial Version:

Referees' comments:

Referee #1 (Remarks to the Author):

This manuscript describes a novel approach to the (proxy) determination of ice shelf thickness variability that enable the record to be extended from the post-1992 timeframe enabled by satellite altimeters to a nearly ~50 year record. This is an important contribution to our understanding of how the ice sheet margin has been changing over time. The results indicate that such pinning point tracking is a useful tool in assessing ice shelf change and the manuscript briefly discusses the implications of further pinning point loss around Antarctica.

In particular, the production of two new near-cloud-free mosaics of Antarctica for both 1973 and 1989 fills a huge need in the community and these mosaics are significant contributions to extending the impact of this work in and of themselves.

This study is a interesting one and uncovers new decade-scale information about the history of change in ice shelf thicknesses prior to satellite altimetry, but in my opinion the manuscript itself has to be modified before it is accepted for publication. While there is no doubt a valuable data set here, the manuscript as it stands feels a little bit incomplete, and its impacts unclear. It comes across as a listing of evidence at trial, but ends before it fully synthesizes all of that evidence into a cohesive story. While I know that Nature's format is short-form, readers are sort of plopped into the middle of the introduction without context for what a pinning point is, given a list of what has changed, and then wrapped up with a statement about implications of further pinning point change leading to mass loss. Again, no doubt that the authors have created a valuable dataset and have interesting results about pinning point change, but I think the manuscript as it stands does not synthesize that information in a way that fully sets the results in context or fully reflects the aims denoted in the introduction.

To the point of lacking context, I think what might set it up this way to begin is that the manuscript does not fully define what "pinning point" means at the outset, instead assuming that readers know what a pinning point is. While it is described in the methods section, it would be more beneficial to articulate what pinning points are up front so that the impact of their loss means something. Non-specialists would have no context clues as to what a pinning point physically looks like or (even after looking at Figure 3) what causes the surface expression shown in Figure 3. Without a definition of what pinning points are, it may be hard for non-specialist readers to interpret results sentences like "reduced in area" and "pinning points were lost" mean physically. In addition, while again it is described (briefly) in the Methods section, it would be beneficial to describe what the results are showing in terms of what was measured; in essence, the authors judged growth, shrinking, or null change by qualitatively comparing how a pinning point looked in the series of satellite images. The

point of the manuscript was presented as a proxy for ice shelf thickness, so how surface area change corresponds to thickness needs to be more obvious. This was nicely shown in the authors' Extended Data Figure 1a and 1b, and I wonder if this might move into one of the main figures at the beginning to both establish what pinning points are and how their surface expression (change in which is being shown in Figs. 1 and 2; examples in Fig 3) actually relates to thinning/thickening would be value added.

To the point of synthesizing information, I think this is generally a difficult task for observation papers like this one. It is tempting (and necessary) to describe observations, but at a certain point it turns into a list and readers are left with "ok, but what does all that mean?" I think the overall impact of the manuscript could be improved if the observations were synthesized better and, more importantly, tied to the goal of extending the ice shelf thickness change record. Because the goal of the manuscript (as described in the text) was to extend the record of ice shelf thickness change over time, it was confusing that none of the figures related to ice shelf thickness. Much of the results section described changes in pinning points, which is fine, as I realize the authors are focused on the pinning points themselves. But all in all, as described in the manuscript, the reason we want to look at pinning point change is because it informs on ice shelf thickness changes and stability, but this connection to the bigger picture was lacking overall (though it was presented as motivation for the study). I just think there is a disconnect between the valuable data the authors have produced with the bigger-picture impacts. Again, it's set up well in the introduction, but the connection between results and implications is not as successful.

As it stands, the manuscript does not fully articulate its impact on state-of-knowledge. The reason this work would have a large impact would be because it extends the record on ice shelf thickness changes. But as it stands, it is not fully framed that way; other than a few places, results were discussed from the pinning point change quantification rather than going one step further to describe what that meant for thinning/thickening and implications for the ice shelves being discussed. While it does discuss overall results (e.g., pinning points have overall reduced in area at an increasing rate since the 1970s), and this is a useful result in terms of extending the record back ~50 years, I'm not sure this is a surprising result, and I expected the more interesting results to come out in the results section where the authors shed light on regional/local signals that tell us more about how/when/why some ice shelves are changing more than others, etc. But this is where the results section turned into a bit of a list rather than interpretations of what the changes (or not) meant for ice shelf stability and/or local environmental changes. For example, the results section lists pinning point growth on Abbott Ice Shelf as "surprising" but then stated it is "unknown if this thickening is connected to the long-term increase in snowfall over the ice shelf or possible long-term reductions in basal melt rates". I imagine this is true for all of the results, but the methods section does not describe ways in which the authors could differentiate between surface-induced (snowfall) or basal melt-induced geometry changes over time, and that would have significant implications for how to interpret this dataset as a whole, particularly if the goal was to elucidate thickness change over time by basal melting, which is the leading cause of thickness changes (particularly over decadal timescales like this one). Because of that, the impact on the state of knowledge was a bit fuzzy. While I found the conclusions to be based on a great new dataset by the authors, the conclusions were not particularly impactful at this stage.^[1]_[SEP]

While I know there is limited figure space, I think there are a few improvements that could be made to help readers synthesize the information presented. Firstly, in Figure 2 (and by association Figure 1) it might be easier for readers to determine the “X” vs “+” symbols if they were filled vs empty circles, or something like that. Also, the caption lists the “+” as showing 1989-2020 change while the figure shows this to mean 1989-2022 change. Second, I would suggest (as before) moving Extended Data Figure 1a and 1b to somewhere in the main figures, as it helps to visualize and contextualize what the authors are describing, particularly for non-specialists.

The supplementary figures were well produced and I had no comments on changes to those. The manuscript itself was well-referenced.

Referee #2 (Remarks to the Author):

Review of Miles & Bingham, Nature, April 2023

Authors made a massive and unprecedented effort to document changes in the pinning points that dynamically affect the flow of Antarctic ice shelves (and ultimately the grounded ice sheet upstream) over a time period of 50 years through visual inspection of novel cloud free Landsat mosaics.

This study came as a surprised to me as I did not know that the “simple” (but certainly time consuming) visual examination of repeat Landsat imagery could reveal at the scale of the entire Antarctic continent the thickness changes of the ice shelves. The potentially important added value of this work is that recent changes (measured quantitatively during the last 30 years from radar altimetry) can be placed now in a 50-yr perspective. For example, decades when rapid changes started can be temporally better constrained. However, the impact of the article would be stronger if the observations could be made more quantitative through further comparison to ice shelf thickness change over their overlapping period (1990s-now).

General comments.

1/ I would have like to see a more systematic and quantitative comparison with measurements from radar altimetry over the overlapping period (1990s to 2020s). That would help to convince the reader that the visual evolution of pinning points are indeed representative of the ice shelves that flow over them. It would somehow help to make the present assessment more quantitative.

2/ Another important comment concerns the subjective nature of the classification. I appreciate the efforts of the authors to provide quicklooks for an extensive selection of pinning points in the supplement. From these images, I found it sometimes difficult and subjective to decide in what direction a pinning point was evolving as the radiometric quality and illumination vary among the images. This is possibly due to the static nature of an article figures, by essence. This could be partly resolved by providing to reviewers/readers subsets of the images so that they can flicker between them (or directly providing animated GIFs) and thus apprehend the changes more easily. Additionally, I wonder whether an independent assessment by multiple persons would make the conclusions more reliable (reproducible). If 5-10 colleagues were asked to assess the evolution of a

selection of 5 to 10 pinning points would they reach the same conclusions? I understand this is a weird test to ask for a two-author study...

3/ Section L64-131 (regional description) is very long and descriptive. This is of course of great interest for the researcher knowing these individual Antarctic glaciers but probably not to the more general Earth Science reader in Nature. Here I wondered whether it would not be more impactful to make the paper even shorter, pushing the glacier-by-glacier description in a supplement.

Specific comments.

L11. Here the reader wonders whether "200" is a lot, or not (maybe add "out of XXX")

L12. Not sure why coastline is used here. Pinning points are inland of the coastline.

L15. Reading the abstract first time, I thought 2000s was used to design the first decade of the 21st century only, while in fact this is for two decades.

L50. Provide the image resolution of the mosaics.

L54. Although I reckon the huge amount of work behind producing an ice shelf Antarctic-wide mosaic, I would have love to see an intermediate date around 2010s (if possible, maybe this corresponds to a gap in the Landsat period due to Scan Line Corrector -SLC- failure). In order to split these 22 years into two time periods at a time when simultaneous radar altimetry data are available for quantitative comparison of the two methods to measure ice shelves thinning (see also general comment #1). Or maybe a mosaic for 2013-14 from Landsat-8 early data?

L57. Before providing results, authors should include a very brief description of the methods, i.e. telling that they have examined the changes in areas (did they draw some polygons for that or, as I understood, just performed visual examination?) and then calculated percentage (by numbers) of pinning point area increase/stable/decrease (I know the methods need to be detailed elsewhere in a Nature paper).

L60. See general comment #1. This is here that one expect a more quantitative analysis of the agreement (or not) between this study and earlier quantitative observations using radar altimetry.

L62. The numbers (15, 25 and 37%) rather suggest a steady increase of the number of pinning points experiencing reduction in area. How do the authors back up a 1990s onset?

L64. I miss here an introductory sentence indicating that authors will examine next regional variations and also telling what geographical logic/order will be followed.

L67 "the" rather than "to"

L69. "Similar" to the Eastern Peninsula? Or the two periods are "similar"? I was unsure.

L107. Do authors mean Peninsula and WAIS here by "other regions"?

L113. Is "erosion" the most appropriate term?

L133. "ice rises". Need to define them like "ice rumples", how do they relate to pinning points describe earlier (is the equation correct: pinning points = ice rise + ice rumple). A justification why they are considered specifically now?

L138. why? And why not between 1973 and 1989?

L157. "a" not needed

Bibliography. titles missing for several articles in the ref list.

Figure 1: what is the explanation behind the increase by number (n) of mapped pinning points? Effect of image quality/resolution?

L371. Provide total number of images downloaded, if available

L408. confusing referencing (4) or (43)?

L418. What about "disappearing" pinning points (included in "decreasing")?

L420. My suggestion would to add animated GIFs to the supplement so that the changes can really be perceived.

L433. What about the 2022 mosaic? Availability?

Author Rebuttals to Initial Comments:

Referee #1 (Remarks to the Author):

This manuscript describes a novel approach to the (proxy) determination of ice shelf thickness variability that enable the record to be extended from the post-1992 timeframe enabled by satellite altimeters to a nearly ~50 year record. This is an important contribution to our understanding of how the ice sheet margin has been changing over time. The results indicate that such pinning point tracking is a useful tool in assessing ice shelf change and the manuscript briefly discusses the implications of further pinning point loss around Antarctica. In particular, the production of two new near-cloud-free mosaics of Antarctica for both 1973 and 1989 fills a huge need in the community and these mosaics are significant contributions to extending the impact of this work in and of themselves.

We thank the reviewer for taking the time to offer feedback on our manuscript. We are delighted that the reviewer finds our work important and novel. We respond to their constructive suggestions below.

This study is a interesting one and uncovers new decade-scale information about the history of change in ice shelf thicknesses prior to satellite altimetry, but in my opinion the manuscript itself has to be modified before it is accepted for publication. While there is no doubt a valuable data set here, the manuscript as it stands feels a little bit incomplete, and its impacts unclear. It comes across as a listing of evidence at trial, but ends before it fully synthesizes all of that evidence into a cohesive story. While I know that Nature's format is short-form, readers are sort of plopped into the middle of the introduction without context for what a pinning point is, given a list of what has changed, and then wrapped up with a statement about implications of further pinning point change leading to mass loss. Again, no doubt that the authors have created a valuable dataset and have interesting results about pinning point change, but I think the manuscript as it stands does not synthesize that information in a way that fully sets the results in context or fully reflects the aims denoted in the introduction.

To the point of lacking context, I think what might set it up this way to begin is that the manuscript does not fully define what "pinning point" means at the outset, instead assuming that readers know what a pinning point is. While it is described in the methods section, it would be more beneficial to articulate what pinning points are up front so that the impact of their loss means something. Non-specialists would have no context clues as to what a pinning point physically looks like or (even after looking at Figure 3) what causes the surface expression shown in Figure 3. Without a definition of what pinning points are, it may be hard for non-specialist readers to interpret results sentences like "reduced in area" and "pinning points were lost" mean physically. In addition, while again it is described (briefly) in the Methods section, it would be beneficial to describe what the results are showing in terms of what was measured; in essence, the authors judged growth, shrinking, or null change by qualitatively comparing how a pinning point looked in the series of satellite images. The point of the manuscript was presented as a proxy for ice shelf thickness, so how surface area change corresponds to thickness needs to be more obvious. This was nicely shown in the authors' Extended Data Figure 1a and 1b, and I wonder if this might move into one of the main figures at the beginning to both establish what pinning points are and how their surface expression (change in which is being shown in Figs. 1 and 2; examples in Fig 3) actually relates to thinning/thickening would be value added.

We thank the reviewer for these constructive suggestions. We have now amended the introduction to define and introduce pinning points, but also clarify why pinning-point change is a proxy for ice-shelf thickness and how we mapped pinning-point change i.e. by three-way classification of "shrinking/growing/no change". We have also moved Extended Data Figure 1 to the main manuscript

as Figure 1. We also include a series of new animated imagery on the basis of Reviewer 2's comments that may also help with the transparency of our underlying method.

To the point of synthesizing information, I think this is generally a difficult task for observation papers like this one. It is tempting (and necessary) to describe observations, but at a certain point it turns into a list and readers are left with "ok, but what does all that mean?" I think the overall impact of the manuscript could be improved if the observations were synthesized better and, more importantly, tied to the goal of extending the ice shelf thickness change record. Because the goal of the manuscript (as described in the text) was to extend the record of ice shelf thickness change over time, it was confusing that none of the figures related to ice shelf thickness. Much of the results section described changes in pinning points, which is fine, as I realize the authors are focused on the pinning points themselves. But all in all, as described in the manuscript, the reason we want to look at pinning point change is because it informs on ice shelf thickness changes and stability, but this connection to the bigger picture was lacking overall (though it was presented as motivation for the study). I just think there is a disconnect between the valuable data the authors have produced with the bigger-picture impacts. Again, it's set up well in the introduction, but the connection between results and implications is not as successful.

As it stands, the manuscript does not fully articulate its impact on state-of-knowledge. The reason this work would have a large impact would be because it extends the record on ice shelf thickness changes. But as it stands, it is not fully framed that way; other than a few places, results were discussed from the pinning point change quantification rather than going one step further to describe what that meant for thinning/thickening and implications for the ice shelves being discussed. While it does discuss overall results (e.g., pinning points have overall reduced in area at an increasing rate since the 1970s), and this is a useful result in terms of extending the record back ~50 years, I'm not sure this is a surprising result, and I expected the more interesting results to come out in the results section where the authors shed light on regional/local signals that tell us more about how/when/why some ice shelves are changing more than others, etc. But this is where the results section turned into a bit of a list rather than interpretations of what the changes (or not) meant for ice shelf stability and/or local environmental changes.

The descriptive nature of our section '50 years of pinning-point change' has also been picked up by reviewer 2 and the editor. As a result we have re-written most of this section.

In our revised version we have made this section more focused on ice-shelf thickness change, instead of a blow by blow account of the percentage pinning-point reductions/growth for most ice shelves. Specifically, on the basis of the suggestion from reviewer 2, we have included a new paragraph at the start of the section that compares our pinning-point change observations to satellite altimetry between 2000-2022 and the ICESat-ICESat2 thickness-change observations from 2003-2019. This has improved the manuscript on two accounts. Firstly, the comparison combined with the theory behind pinning-point change (Figure 1 – formerly Extended Data Figure 1) provides validation that our observations in the 1970s-1990s do reflect the spatial pattern of ice-shelf thickness change. But it also makes it clear early on in the results section of the manuscript that while we do measure pinning points, our primary goal in doing this is to extend the ice-shelf thickness change record.

We have also added two new figures comparing ice-shelf thickness change observations from satellite altimetry and our pinning-point change observations, which again we feel helps to focus in our primary goal of extending the record of ice-shelf thickness change.

Throughout this wider section ('50 years of pinning-point change') we have also striven to make our results less descriptive in terms of pinning-point quantification and incorporated a greater discussion of the implications of the thickening/thinning of ice shelves we observe.

We have also re-written the abstract with a greater focus on ice shelf thickness change.

For example, the results section lists pinning point growth on Abbott Ice Shelf as “surprising” but then stated it is “unknown if this thickening is connected to the long-term increase in snowfall over the ice shelf or possible long-term reductions in basal melt rates”. I imagine this is true for all of the results, but the methods section does not describe ways in which the authors could differentiate between surface-induced (snowfall) or basal melt-induced geometry changes over time, and that would have significant implications for how to interpret this dataset as a whole, particularly if the goal was to elucidate thickness change over time by basal melting, which is the leading cause of thickness changes (particularly over decadal timescales like this one). Because of that, the impact on the state of knowledge was a bit fuzzy. While I found the conclusions to be based on a great new dataset by the authors, the conclusions were not particularly impactful at this stage.

We certainly agree with the reviewer that ocean-driven basal melting is the leading cause of thickness change over decadal timescales. Specifically, for Abbot Ice Shelf we have re-framed our discussion. We now treat Abbot Ice Shelf in a regional Bellingshausen Sea context. All ice shelves in this sector thinned over the course of our observations, except for Abbot and Wilkins ice shelves. We explain this by the fact that both Abbot and Wilkins ice shelves are relatively thin, meaning the bases of these ice shelves are less likely to be in regular contact with warmer waters at the bottom of the ocean column.

While I know there is limited figure space, I think there are a few improvements that could be made to help readers synthesize the information presented. Firstly, in Figure 2 (and by association Figure 1) it might be easier for readers to determine the “X” vs “+” symbols if they were filled vs empty circles, or something like that. Also, the caption lists the “+” as showing 1989-2020 change while the figure shows this to mean 1989-2022 change. Second, I would suggest (as before) moving Extended Data Figure 1a and 1b to somewhere in the main figures, as it helps to visualize and contextualize what the authors are describing, particularly for non-specialists.

We have changed the “+” symbol to “” to make it easier for the reader to determine the differences between the symbols. We note filled vs empty circles had a similar problem in that the empty circles converged to look like filled circles in regions where there are many pinning points close together.*

We have also amended the caption and moved Extended Data Figure 1 into the main manuscript as a new Figure 1.

The supplementary figures were well produced and I had no comments on changes to those. The manuscript itself was well-referenced.

Thank-you

Referee #2 (Remarks to the Author):

Authors made a massive and unprecedented effort to document changes in the pinning points that dynamically affect the flow of Antarctic ice shelves (and ultimately the grounded ice sheet upstream) over a time period of 50 years through visual inspection of novel cloud free Landsat mosaics.

This study came as a surprise to me as I did not know that the “simple” (but certainly time consuming) visual examination of repeat Landsat imagery could reveal at the scale of the entire Antarctic continent the thickness changes of the ice shelves. The potentially important added value of this work is that recent changes (measured quantitatively during the last 30 years from radar altimetry) can be placed now in a 50-yr perspective. For example, decades when rapid changes started can be temporally better constrained. However, the impact of the article would be stronger if the observations could be made more quantitative through further comparison to ice shelf thickness change over their overlapping period (1990s-now).

We thank the reviewer for taking the time to provide feedback on our manuscript. We are delighted that the reviewer has found our study interesting and novel. We have responded to their constructive suggestions below.

General comments.

1/ I would have like to see a more systematic and quantitative comparison with measurements from radar altimetry over the overlapping period (1990s to 2020s). That would help to convince the reader that the visual evolution of pinning points are indeed representative of the ice shelves that flow over them. It would somehow help to make the present assessment more quantitative.

We agree that a quantitative comparison to satellite altimetry over the overlapping period improves this manuscript and we now incorporate this into the manuscript . Indeed, the exercise has shown that in places a careful examination of changes to pinning points and grounding lines may serve to validate inferences drawn from satellite altimetry. Satellite altimetry of course has the advantage in that it can directly quantify thickness change over gridded/continuous coverages, but it is known to struggle in some of the narrower ice shelves or those with rougher topography where sampling is more limited. Indeed, we show this in our revised version for the case of George VI Ice Shelf.

In addition to a new Paragraph 2 in the section “50 years of pinning-point change” devoted to this comparison exercise, we have added two new Extended-Data figures. Extended Data Figure 2 presents a side by side comparison between pinning-point change from 2000-2022 and altimetry-measured ice-shelf thickness change from 2003-2019. Extended Data Figure 3 highlights some local anomalies where satellite altimetry shows ice shelves as thickening, yet extensive pinning-point loss and grounding-line retreat strongly suggest that satellite altimetry is not capturing the true direction of thickness change in these isolated locations.

2/ Another important comment concerns the subjective nature of the classification. I appreciate the efforts of the authors to provide quicklooks for an extensive selection of pinning points in the supplement. From these images, I found it sometimes difficult and subjective to decide in what direction a pinning point was evolving as the radiometric quality and illumination vary among the images. This is possibly due to the static nature of an article figures, by essence. This could be partly resolved by providing to reviewers/readers subsets of the images so that they can flicker between

them (or directly providing animated GIFs) and thus apprehend the changes more easily. Additionally, I wonder whether an independent assessment by multiple persons would make the conclusions more reliable (reproducible). If 5-10 colleagues were asked to assess the evolution of a selection of 5 to 10 pinning points would they reach the same conclusions? I understand this is a weird test to ask for a two-author study...

We agree that the changes in the surface expression of pinning points are more readily visualized when images are flicked between each other, as a complement to static side-by-side comparisons. To address this we have produced a number of GIFs covering prominent ice shelves. We feel these are a vast visual improvement and hope the provision of these enables the reviewer and others to apprehend the changes more easily. They also have an added benefit in enables readers to visualize the structural changes of ice shelves, potentially extending the impact of this study.

We do not feel that an independent assessment by multiple authors would make the dataset more reliable. Our justification for this is that asking another author/colleague to map the change with no or limited experience in handling the Landsat imagery or viewing pinning points is not a like for like comparison with an author who has handled 100s of early Landsat images and viewed 1000+ pinning points over varying time periods.

3/ Section L64-131 (regional description) is very long and descriptive. This is of course of great interest for the researcher knowing these individual Antarctic glaciers but probably not to the more general Earth Science reader in Nature. Here I wondered whether it would not be more impactful to make the paper even shorter, pushing the glacier-by-glacier description in a supplement.

This has also been picked up by reviewer 1 and the editor and we have re-written most of this section.

We think it is important to provide a circum-Antarctic view of the changes rather than to focus too much on the typical regions of change e.g. Amundsen Sea and Antarctic Peninsula. The point here is that even highlighting regions with no change is important because of the extended length of our dataset. That said, we agree that the previous format was too descriptive and probably would not be as interesting for a general Earth Science reader. Thus we have:

- *Removed some of the blow by blow account of each individual ice-shelf %age pinning-point change*
- *Transferred some of the discussion previously in the 'bleak future for some ice shelves' section into the regional trends section i.e. Confirmation that ice-shelf thinning started in the Amundsen Sea sector at least in the early 20th century.*
- *Added new discussions of the processes driving the changes in ice-shelf thickness and the wider implications*

Specific comments.

L11. Here the reader wonders whether "200" is a lot, or not (maybe add "out of XXX")

We have re-written the abstract to focus more on ice-shelf thickness change and this phrase is no longer used.

L12. Not sure why coastline is used here. Pinning points are inland of the coastline.

We have re-written the abstract to focus more on ice-shelf thickness change and this phrase is no longer used.

L15. Reading the abstract first time, I thought 2000s was used to design the first decade of the 21st century only, while in fact this is for two decades.

We have re-written the abstract to focus more on ice-shelf thickness change and this phrase is no longer used.

L50. Provide the image resolution of the mosaics.

We have added the spatial resolution of the mosaics (60 m for 1973 and 30 m for 1989).

L54. Although I reckon the huge amount of work behind producing an ice shelf Antarctic-wide mosaic, I would have love to see an intermediate date around 2010s (if possible, maybe this corresponds to a gap in the Landsat period due to Scan Line Corrector -SLC- failure). In order to split these 22 years into two time periods at a time when simultaneous radar altimetry data are available for quantitative comparison of the two methods to measure ice shelves thinning (see also general comment #1). Or maybe a mosaic for 2013-14 from Landsat-8 early data?

The reviewer is correct in that the SLC failure of Landsat-7 inhibits any possibility of a 2010 mosaic that covers all ice shelves. Landsat-8 has regular coverage of Antarctica from mid November 2013 onwards, meaning there is only partial coverage of the austral summer 2013/2014. Therefore, because of cloud cover, we would have to include a lot of imagery from austral summer 2014/2015 as well. Therefore because of image availability we would only be able to produce datasets from 2000 - 2014/2015 and 2014/2015 – 2021/2022. The shortness of the time gap between the second epoch (potentially just 6 years) would not be a like for like comparison. In just 6 years any change in the surface expression of pinning points is likely to be very small. This method is far more suitable over longer time periods.

L57. Before providing results, authors should include a very brief description of the methods, i.e. telling that they have examined the changes in areas (did they draw some polygons for that or, as I understood, just performed visual examination?) and then calculated percentage (by numbers) of pinning point area increase/stable/decrease (I know the methods need to be detailed elsewhere in a Nature paper).

We have added this short description of the methods before the results.

L60. See general comment #1. This is here that one expect a more quantitative analysis of the agreement (or not) between this study and earlier quantitative observations using radar altimetry.

We have added a new paragraph here with a more quantitative analysis. Please see comment above.

L62. The numbers (15, 25 and 37%) rather suggest a steady increase of the number of pinning points experiencing reduction in area. How do the authors back up a 1990s onset?

We have slightly rewritten this paragraph and the “1990s onset” sentence has been removed.

L64. I miss here an introductory sentence indicating that authors will examine next regional variations and also telling what geographical logic/order will be followed.

We have added this sentence: "In the following sections, we focus on the regional variations in pinning point change across the Antarctic Peninsula, West Antarctica and East Antarctica."

L67 "the" rather than "to"

Amended.

L69. "Similar" to the Eastern Peninsula? Or the two periods are "similar"? I was unsure.

Phrase has been removed.

L107. Do authors mean Peninsula and WAIS here by "other regions"?

Yes. We have now clarified this in the text: "However, unlike for West Antarctica and the Antarctic Peninsula, there has been no clear acceleration in the proportion of pinning points reducing in area."

L113. Is "erosion" the most appropriate term?

Amended to "melting".

L133. "ice rises". Need to define them like "ice rumples", how do they relate to pinning points describe earlier (is the equation correct: pinning points = ice rise + ice rumple). A justification why they are considered specifically now?

We have included a definition of ice rises in the opening sentence of the paragraph. We justify their inclusion here because they are fundamental to former ice-sheet reconstruction but, to the best of our knowledge, there have been no remotely-sensed observations that have highlighted that they can change quite significantly over decades.

L138. why? And why not between 1973 and 1989?

The ice rise must have ungrounded early on in the time period 1973-1989. This is because by 1989 it had already migrated some distance from its original position (See Fig. 4a). We have amended the text to make this more clear: "The 5 km-wide Borchgrevink Ice Rise ungrounded in the late 1970s (Fig. 4a), despite expressing limited ice-shelf thickness change in modern satellite-altimetry records. This hints at vigorous ice-shelf thinning occurring prior to its ungrounding the late 1970s"

L157. "a" not needed

Amended.

Bibliography. titles missing for several articles in the ref list.

Amended

Figure 1: what is the explanation behind the increase by number (n) of mapped pinning points? Effect of image quality/resolution?

This is largely a factor of imager availability. In the 1970s mosaic we have no coverage of the Northern Peninsula (Larsen A, B,C, Wordie). This explains a large part of the increase in pinning points because there are many pinning points located in these regions. There are also a few spots of cloud in both the 1973 and 1989 mosaics.

L371. Provide total number of images downloaded, if available

The numbers of images included in the 1973 and 1989 mosaics are stated later in the paragraph. We did not record the total number of images downloaded (i.e. those scenes that were downloaded but not included in the mosaic.

L408. confusing referencing (4) or (43)?

We have amended the citation to read: "In addition, we used the MEaSURES interferometry grounding-line product(49,50) to detect pinning points that were not included in (6)."

L418. What about "disappearing" pinning points (included in "decreasing")?

We confirm that disappearing pinning points are included in "decreasing in extent". We have amended the text to clarify this.

L420. My suggestion would to add animated GIFs to the supplement so that the changes can really be perceived.

We have added these. Please see comment above.

L433. What about the 2022 mosaic? Availability?

The 2022 mosaic is in the process of being uploaded to the Edinburgh Data Share server. A link will be available for those interested to download in a similar manor to the Landsat-1 and Landsat-4 mosaics (<https://datashare.ed.ac.uk/handle/10283/4801>), but this may take a couple of weeks to generate.

Reviewer Reports on the First Revision:

Referees' comments:

Referee #1 (Remarks to the Author):

The authors have improved the manuscript since I last read it in particular by defining and contextualizing pinning points for the reader at the start. The authors also improved its readability by streamlining results, describing overarching interpretations rather than listing items without connection to one other, as they appeared in the first version. The study remains an interesting and novel one and I think its conclusions are broadly appropriate and of interest to the geoscience community. I have one remaining broader comment about consistency of approach and necessity of explaining that; and a few minor comments, all below.

Broadly, I'm not sure I follow the reasoning behind the weaker correlation between pinning points that reduced in extent in imagery and regions of altimetry-detected ice-shelf thinning. I think it highlights a bigger comment that I have that relates to the method overall - ways in which lighting conditions were accommodated in the main task of characterization of pinning point change in itself was not well described. I think that the section describing validation against altimetry data highlights this limitation in particular. The authors describe their approach using visible imagery ("zooming into each pinning point, finding the optimum contrast,...flicking between each successive epoch"), but do not describe a consistent image processing approach to accommodating lighting changes aside from "finding optimum contrast". I am not sure that a very rigorous application is needed, but certainly it should be more explicitly acknowledged that lighting conditions (e.g., sun angle) have a significant impact on the visibility of low-contrast features (e.g., Antarctic ice shelves where almost everything is bright), particularly when the sun is directly/close to overhead in high summer vs high sun angles producing large contrasts/shadows closer to fall/spring times. The authors *do* state that in they "erred on the side of caution and classified ["cases where it is unclear if the surface expression of pinning points has changed"] as showing no detectable change", but the manuscript right now actually does not explain how differences in illumination conditions were handled in the assembly of the mosaics, which suggests that different sections of the mosaics can represent very different lighting conditions but are being categorized as the same, representative of an entire time period rather than a snapshot in time. Overall, I think the approach is fine and I think the method yields a useful and novel result so I am not suggesting the authors have to re-do any analyses. However, I think these points should be a lot more clear, particularly when validating the results against the altimetry records.

The authors argue that the weaker correlation between pinning points that reduced in extent in imagery and regions of altimetry-detected ice-shelf thinning is due to "localised instances" primarily on the AP; they suggest that *visible* grounding line retreat and the magnitude by which the pinning point shapes changed indicate that satellite altimetry observations of ice-shelf thickness change is wrong in these areas. I really think the balance between satellite altimetric biases and image biases has to be better struck here. First, I wonder how the grounding line retreat was measured (the red lines shown in Ext Fig 3; the main text Line 81-82 suggests it came from visible imagery assessments, but the extended material, Line 511, suggests the MEaSURES GL positions are used). The accuracy of the ICESat-ICESat-2 thickness change dataset comes from years of repeat-

track detections (while that doesn't completely account for all issues with those data sets, it certainly is more robust to cloud issues than is suggested here), and is independent of lighting conditions during winter or summer; the same cannot be said for the approach used here. And so that's all a long way of saying - I don't think the equivalency of laser altimetry accuracy with variable contrast imagery is quite appropriate. It might just be that an odd sun angle made a pinning point look smaller at the exact time the clearest image occurred. Just couch the mis-match a little more accurately, I suppose. Or, I would suggest at least discussing a little more about the consistency of the approach with regard to sun angle in imagery (both for judging pinning point surface change and for GL extraction, if imagery was used for that) and how that might impact the results.

Line 9-10: I would suggest changing "...and we do not know which ice shelves were 10 thinning before then" to "and as such information about prior thinning behaviors remains unquantified" or something like that, to reflect more on the need for quantification of thinning patterns (as is enabled by the altimetry referenced in the previous sentence) rather than just "knowing" which ones were thinning.

Line 54: "For all of these reasons understanding" needs a comma - "For all of these reasons, understanding"

Lines 130-132: The authors state "Collectively these patterns imply a decadal-scale raising of the thermocline depth and thickening of the layer of warm water on the continental shelf across the entire Bellingshausen Sea sector since 2000" - is this the case? Are there studies that corroborate this finding? I might be wrong but I think some modeling of water column change in this region exists and could be cited here as an extra validation of these results matching what others are seeing in the ocean...

Line 148: "spreadbetween" needs a space

The new Figure 1 is a great improvement to the manuscript that demonstrates the relatively simple but time-consuming (!!) method applied by the authors without adding too much text. The only change I would make to this figure is circling the same spots on both before and after images in c and d, because the different lighting conditions might make it difficult for folks who are not used to looking at imagery like this to lock onto what they're comparing between images. (Obviously it's the lack of features that the authors are trying to point out by having no circles in the thinner cases in c (lower) and d (upper), but I think having perhaps dotted circles around the same area might make that point more obvious?)

Referee #2 (Remarks to the Author):

Re-Review of Miles & Bingham, Nature, August 2023

Authors carefully examined all suggestions from the referees and this led to an improved and more quantitative study with important implications for the decadal stability of the Antarctic ice shelves. I still have a major comment and several others smaller points.

General comments.

1/ The addition of ice shelf elevation changes derived from laser altimetry is welcome. However, the way the comparison is presented is not sound (or not clear). In Extended Data Figure 2, the three categories to separate different pinning point behaviours are based on +0.5 m/yr and -0.5 m/yr thresholds on the rate of elevation change. However in the text, the threshold is different (1 m/yr) and it seems that the categories overlap (the text L543-547 is not clear and suggests a strong overlap of categories), i.e. some pinning points are considered has both stable and reducing (or stable and increasing) which, if true, is not appropriate as such an overlap does not exist in the classification based in the visual inspection of pinning points. My suggestion is to use three categories of thinning rate and report, in a table, the number (and percentage) of increasing/stable/decreasing pinning points for each category of dh/dt (and summarize the main values in the text). Possible categories (as in Ext Fig. 2) <-0.5 m/yr ; $[-0.5 ; +0.5]$; > 0.5 m/yr but authors may want to adjust the thresholds based on the uncertainties on the laser altimetry product (when a thinning rate is unambiguously positive/negative?).

2/ Author had a fair justification for not using radar altimetry data. However, in the cases (for example those in Extended Data Figure 3) where laser altimetry and visual inspection of Landsat imagery led to conflicting evolutions, it would be good to check the radar altimetry data time series. It would nourish further the discussion and possibly help to back up the authors' statement that laser altimetry is likely erroneous in these specific cases.

Specific comments.

L20. This statement regarding the future behaviour at the end of the abstract is more "speculation" than solid knowledge gained from the study. Replace "will" by "would" or write something like "This trend, if continuing, will further" to underline the uncertain nature of this statement.

L75. what is "limited changes"? Authors need to be more specific already here in the article about the thresholds applied to dh/dt from altimetry to define these categories.

L80. "Magnitude". Is it high (I guess) or low ? Right now not 100% clear.

L102-103. I do not think the authors need to recall the agreement. It was clearly outlined already.

L109. "Ice rise" is defined only L208. Move definition here?

L125. This is consistent “with”

L133. This statement would be more meaningful if % of stable/growing pinning points were also indicated. e.g., if 65% (the rest) of the pinning points were growing then the implications would be very different (I guess that, in fact, most of the rest are stable).

L148. space missing

L513. “either dataset”. I did not really understand what where the two datasets. the one from ref 6. Which is the other one?

Author Rebuttals to First Revision:

Referees' comments:

Referee #1 (Remarks to the Author):

The authors have improved the manuscript since I last read it in particular by defining and contextualizing pinning points for the reader at the start. The authors also improved its readability by streamlining results, describing overarching interpretations rather than listing items without connection to one other, as they appeared in the first version. The study remains an interesting and novel one and I think its conclusions are broadly appropriate and of interest to the geoscience community. I have one remaining broader comment about consistency of approach and necessity of explaining that; and a few minor comments, all below.

We thank the reviewer for again taking the time to provide thoughtful and constructive comments.

Broadly, I'm not sure I follow the reasoning behind the weaker correlation between pinning points that reduced in extent in imagery and regions of altimetry-detected ice-shelf thinning. I think it highlights a bigger comment that I have that relates to the method overall - ways in which lighting conditions were accommodated in the main task of characterization of pinning point change in itself was not well described. I think that the section describing validation against altimetry data highlights this limitation in particular. The authors describe their approach using visible imagery ("zooming into each pinning point, finding the optimum contrast,...flicking between each successive epoch"), but do not describe a consistent image processing approach to accommodating lighting changes aside from "finding optimum contrast". I am not sure that a very rigorous application is needed, but certainly it should be more explicitly acknowledged that lighting conditions (e.g., sun angle) have a significant impact on the visibility of low-contrast features (e.g., Antarctic ice shelves where almost everything is bright), particularly when the sun is directly/close to overhead in high summer vs high sun angles producing large contrasts/shadows closer to fall/spring times. The authors *do* state that in they "erred on the side of caution and classified ["cases where it is unclear if the surface expression of pinning points has changed"] as showing no detectable change", but the manuscript right now actually does not explain how differences in illumination conditions were handled in the assembly of the mosaics, which suggests that different sections of the mosaics can represent very different lighting conditions but are being categorized as the same, representative of an entire time period rather than a snapshot in time. Overall, I think the approach is fine and I think the method yields a useful and novel result so I am not suggesting the authors have to re-do any analyses.

The issue here is that for large parts of the Antarctic coastline in 1973 and in 1989 there was often only one available cloud free image. This made designing a methodology with a consistent approach in terms of only selecting imagery with a certain solar angle impossible. However, we are confident that this had a very limited effect on our classifications. In the most part pinning point change is obvious, as shown in the series of animated imagery we have provided. To improve the manuscript, we have highlighted the absence of a consistent approach in terms of sun azimuth in the methods.

However, I think these points should be a lot more clear, particularly when validating the results against the altimetry records. The authors argue that the weaker correlation between pinning points that reduced in extent in imagery and regions of altimetry-detected ice-shelf thinning is due to "localised instances" primarily on the AP; they suggest that *visible* grounding line retreat and the magnitude by which the pinning point shapes changed indicate that satellite altimetry observations of ice-shelf thickness change is wrong in these areas. I really think the balance between satellite altimetric biases and image biases has to be better struck here. First, I wonder how the grounding line retreat was measured (the red lines shown in Ext Fig 3; the main text Line 81-82 suggests it came from visible imagery assessments, but the extended material, Line 511, suggests the MEaSURES GL positions are used). The accuracy of the ICESat-ICESat-2 thickness change dataset comes from years of repeat-track detections (while that doesn't completely account for all issues

with those data sets, it certainly is more robust to cloud issues than is suggested here), and is independent of lighting conditions during winter or summer; the same cannot be said for the approach used here. And so that's all a long way of saying - I don't think the equivalency of laser altimetry accuracy with variable contrast imagery is quite appropriate. It might just be that an odd sun angle made a pinning point look smaller at the exact time the clearest image occurred. Just couch the mis-match a little more accurately, I suppose. Or, I would suggest at least discussing a little more about the consistency of the approach with regard to sun angle in imagery (both for judging pinning point surface change and for GL extraction, if imagery was used for that) and how that might impact the results.

We do agree that a comparison between pinning point change and altimetry is not perfect. On reflection we can see that the text could be interpreted that we are claiming that there is a large mismatch between IceSat altimetry and pinning point change and that the altimetry must be wrong in all of these locations. This was not our intention and we appreciate that the text needs to be more nuanced. In the revised text we highlight that this is not a perfect comparison because:

- *For example, when pinning points unground, there is sometimes a localised thickening of the ice shelf downstream. This occurs when ice flow was previously, at least, partly diverted around former pinning points, but then flows uninhibited following ungrounding. Examples of this processes are shown at Crosson (Extended Data Figure 6c) and Borchgrevink (Fig. 3a) ice shelves and may explain some of the mismatches.*
- *Our cautious approach in mapping. We only classifying pinning points where we are certain of change, so some of the pinning points where we classify 'no detectable change' may actually be getting smaller, its just that we cannot clearly see it from the Landsat imagery.*

That said, in very rare cases, for example in Extended Data Figure 4a, we maintain that clear grounding retreat is incompatible with adjacent ice shelf thickening indicated by altimetry. To clarify, we observe the grounding line retreat visually and not from InSAR or the MEASURES product. The grounding line retreat and extensive pinning point loss in this example are very clear and they are definitely not artefacts associated with changing lighting conditions. Indeed, this retreat is even visible on the much coarser MODIS imagery (250 m spatial resolution vrs 15 m for Landsat), as such we invite the reviewer to view this change here on NASA WorldView.

Line 9-10: I would suggest changing "...and we do not know which ice shelves were thinning before then" to "and as such information about prior thinning behaviors remains unquantified" or something like that, to reflect more on the need for quantification of thinning patterns (as is enabled by the altimetry referenced in the previous sentence) rather than just "knowing" which ones were thinning.

We have amended this sentence to "However, observations of ice-shelf thickness change by satellite altimetry only stretch back to 1992 and prior information about thinning remains unquantified"

Line 54: "For all of these reasons understanding" needs a comma - "For all of these reasons, understanding"

Amended

Lines 130-132: The authors state "Collectively these patterns imply a decadal-scale raising of the thermocline depth and thickening of the layer of warm water on the continental shelf across the entire Bellingshausen Sea sector since 2000" - is this the case? Are there studies that corroborate this finding? I might be wrong but I think some modeling of water column change in this region

exists and could be cited here as an extra validation of these results matching what others are seeing in the ocean...

This is a good spot, we missed this paper. Oelerich et al (2022): Wind-Induced Variability of Warm Water on the Southern Bellingshausen Sea Continental Shelf, JGR Oceans. They use ocean reanalysis products to show a warm ocean conditions in the Bellingshausen Sea at ~2007-2017 and cool ocean conditions ~1997-2006. This is reasonably consistent with our results and we cite this paper in the revised version.

Line 148: "spreadbetween" needs a space

Amended

The new Figure 1 is a great improvement to the manuscript that demonstrates the relatively simple but time-consuming (!!) method applied by the authors without adding too much text. The only change I would make to this figure is circling the same spots on both before and after images in c and d, because the different lighting conditions might make it difficult for folks who are not used to looking at imagery like this to lock onto what they're comparing between images. (Obviously it's the lack of features that the authors are trying to point out by having no circles in the thinner cases in c (lower) and d (upper), but I think having perhaps dotted circles around the same area might make that point more obvious?)

This is a good suggestion and we have added the circles to the figure to guide the reader.

Referee #2 (Remarks to the Author):

Authors carefully examined all suggestions from the referees and this led to an improved and more quantitative study with important implications for the decadal stability of the Antarctic ice shelves. I still have a major comment and several others smaller points.

We thank the reviewer for again taking the time to provide thoughtful and constructive comments.

General comments.

1/ The addition of ice shelf elevation changes derived from laser altimetry is welcome. However, the way the comparison is presented is not sound (or not clear). In Extended Data Figure 2, the three categories to separate different pinning point behaviours are based on +0.5 m/yr and -0.5 m/yr thresholds on the rate of elevation change. However in the text, the threshold is different (1 m/yr) and it seems that the categories overlap (the text L543-547 is not clear and suggests a strong overlap of categories), i.e. some pinning points are considered has both stable and reducing (or stable and increasing) which, if true, is not appropriate as such an overlap does not exist in the classification based in the visual inspection of pinning points. My suggestion is to use three categories of thinning rate and report, in a table, the number (and percentage) of increasing/stable/decreasing pinning points for each category of dh/dt (and summarize the main values in the text). Possible categories (as in Ext Fig. 2) <-0.5 m/yr ; $[-0.5 ; +0.5]$; > 0.5 m/yr but authors may want to adjust the thresholds based on the uncertainties on the laser altimetry product (when a thinning rate is unambiguously positive/negative?).

We concur from this comment that our writing and representation of what we have done here were not clearly presented.

One issue pointed out by the reviewer was that in ED Fig. 2 we partitioned the altimetry-detected ice-shelf thickness changes as <-0.5 m/yr ; $[-0.5 ; +0.5]$; > 0.5 m/yr, but in the text, the bins used to quantify the relationship between ice shelf thickness change and mapped pinning point change were <-0 m/yr ; $[-1 ; +1]$; > 0 m/yr used. The purpose of this figure is simply to provide a visual comparison between mapped pinning point change and ice shelf altimetry. Therefore, to avoid any unnecessary confusion we simply revert the colour scale to a standard stretched colour bar.

Regarding the bins used in our quantitative comparison to pinning point change, we maintain that it is important that the middle category overlaps. Theoretically any pinning point where nearby ice shelf thickness change is negative (<0 m yr) should reduce in extent, while any pinning point where nearby thickness change is positive (>0 m yr) should increase in extent. But there are always going to be some pinning points where we cannot observe any change, and the changes in ice shelf thickness here should be close to zero. Therefore, a middle overlapping category is required, where we use the values of between -1 m yr and $+1$ m yr. We think this is now better explained and visualised in the new ED Fig. 3, that plots elevation change we extract from IceSat altimetry for each pinning point, that is colour coded in relation to the direction of pinning point change (smaller, no detectable change, larger).

2/ Author had a fair justification for not using radar altimetry data. However, in the cases (for example those in Extended Data Figure 3) where laser altimetry and visual inspection of Landsat imagery led to conflicting evolutions, it would be good to check the radar altimetry data time series. It would nourish further the discussion and possibly help to back up the authors' statement that laser altimetry is likely erroneous in these specific cases.

This is a good suggestion, we have added this to Extended Data Figure 3. The direction of change is indeed different between the two products and highlights, that at least in highly localised regions,

there are some discrepancies between different altimetry products.

Specific comments.

L20. This statement regarding the future behaviour at the end of the abstract is more “speculation” than solid knowledge gained from the study. Replace “will” by “would” or write something like “This trend, if continuing, will further” to underline the uncertain nature of this statement.

We have replaced “will” with “would”

L75. what is “limited changes”? Authors need to be more specific already here in the article about the thresholds applied to dt/dt from altimetry to define these categories.

We clarify as “between -1 and 1 m yr⁻¹”

L80. "Magnitude". Is it high (I guess) or low ? Right now not 100% clear.

This sentence has been removed.

L102-103. I do not think the authors need to recall the agreement. It was clearly outlined already.

We have removed this duplication.

L109. "Ice rise" is defined only L208. Move definition here?

We have moved the definition to L109.

L125. This is consistent “with”

Amended

L133. This statement would be more meaningful if % of stable/growing pinning points were also indicated. e.g., if 65% (the rest) of the pinning points were growing then the implications would be very different (I guess that, in fact, most of the rest are stable).

15% of pinning points increased in area during this time period (1973-1989), we have added this to the manuscript.

L148. space missing

Amended

L513. “either dataset”. I did not really understand what where the two datasets. the one from ref 6. Which is the other one?

We have amended the text to clarify “either the ice rise and rumples dataset or the MEaSURES product”

Reviewer Reports on the Second Revision:

Referees' comments:

Referee #1 (Remarks to the Author):

I think the discussion of the comparison against altimetry has improved. My points in the last review were meant to pick on the the equating of rigor of the methods (highly accurate, but not error-free laser altimetry vs non-optimized imagery, which understandably is tricky when you have one/two clear images a year and no control over sun-angle and other factors) rather than questioning why altimetric data might see things differently, but from the response to referees and the manuscript adjustments, my comments may have been interpreted to suggest that the altimetry results needed more explanation. That wasn't entirely my intent, but the authors now include some theoretical considerations that I think fit well here anyway, so the additions are well taken.

I think overall the manuscript has improved and provides useful results that are of general interest to the scientific community and specific interest to the glaciological community overall. However, I do have some concerns about the over-interpretability of limited observations, which is not a fault of the work specifically, it's just a fact that we don't have sufficient observations over long timescales. These authors have maximized the use of what does exist to interpret Antarctic ice shelf thinning over three ~decadal epochs, which extends our understanding of how this has changed over the two decades prior to altimetry data becoming available. This is valuable. However, I would caution against over-stating the results as being a record of change over 50 years, which is the message that comes across in this manuscript. What it does do is provide ~decadal snapshots of change, the latter two of which (1989-2000 and 2000-2022) corroborate existing observations of ice shelf thinning measured by altimeters (~1992-onwards). Perhaps a subtle point and I do not mean it to take away from the impact of this work, but I do think it's important to ensure that interpretations have some guardrails which are a bit buried in the text for now. This is particularly exemplified in the title, which suggests that ice shelves have been unanchoring from pinning points for the last 50 years, however a visual inspection of the resultant dataset in Figure 2 (and reading of the descriptive text) highlights that aside from the west Antarctic Peninsula, segments of the Amundsen Sea Sector, and Totten area, much of the ice shelf area around the continent either showed no decisive change or thickened in the first epoch (1973-1989). It is only in the second and third epochs (1989-2000 and 2000-2022) that unanchoring/thinning becomes more widespread, and only in the most recent epoch that I would judge unanchoring to be the predominant observation. If we had data to compare from the 1960s to show unanchoring increased in the 1970s, then we could state that the ice shelves have been unanchoring for 50 years. But that data does not exist, and as such, this methodology can only say for certain that unanchoring increased between the 1973-1989 period and the 1989-2000 period (and further into the 2000-2022 period) - that's more like a ~30 year inference rather than a 50 year inference (i.e., change over the 1973-1989 period do not necessarily mean that the changes occurred in 1973 - they could have occurred in 1988). The fact that ice shelves have been thinning since first measured in ~1992 has already been established with altimeter measurements. This is definitely not intended to say that altimeters are the be-all, end-all, but certainly this has been observed previously. In that way I think the title is exemplary of the way in which this study somewhat overstates its interpretability. I don't disagree that we should sound

the alarm on these decadal scale changes and I think the effort on behalf of these authors is very worthwhile, but we are also in a time at which we don't need to be overstating impacts because that inevitably makes it harder to communicate with clarity.

Specific comments:

Editorial note, I was trained under the notion that abstracts should not have references within them as they are a summary of the work to be discussed. This abstract has references, I assume perhaps Nature has different formatting guidelines?

Line 28: "attributed to warming ocean currents" - Perhaps a semantic point, but it's not the ocean currents that are warming. It's the subsurface bodies of water carried by the currents. I also might be relevant in this area to mention that the majority of ocean-warming-induced acceleration of ice discharge has been in West Antarctica (of course with a few examples in EAIS), to place this statement in context with observations given the interest in WAIS change, and to set up the background and contrast for what the new findings in this manuscript show in terms of Antarctic-wide change eg in Lines 100-102.

Line 71: "ICESat-1" should be just "ICESat"

Line 190-192: I think the language specifically here and generally in the manuscript is a bit over-stated. The results described in this manuscript relate to changes between three epochs (1973-1989; 1989-2000; 2000-2022). Stating that the results show some ice shelves were "already thinning in the 1970s" is not entirely representative of what the methods employed here can show. Because of the snapshot-like approach dictated by the (understandably) limited data in the 70s and 80s (and 90s), we can say that thinning occurred sometime between 1973 and 1989. Technically, the majority of the intervening years are in the 1980s. I am not sure how it can be said with certainty that thinning began in the 1970s; I don't disagree that it's likely, but that's not what this method demonstrates. In all, I don't think this statement is wrong, but I do think it's misleading, and further interpretation of that to mean that triggers existed prior to the 1970s might be a stretch. I think this needs to be couched a little bit more in terms of what interpretations the methods actually allow for vs assuming a change between 1973-1989 was due to changes in the 1970s. This goes more broadly for the manuscript as a whole, as well - while it does enable a ~50-year lookback that hasn't previously been investigated, the methods presented enable an inter-epochal comparison over a 50-year period, rather than a 50-year time series of change, in essence adding two ~decadal-scale periods of information (1973-1989; 1989-2000) to the existing 30 year annual/sub-annual time series record. I think this point is somewhat muddled throughout, including in the title and abstract. I don't mean for this to be a critique of the method or the results, as they are valuable; it's just the language is a bit over-stated in terms of what we can actually interpret from decadal-scale change observations (for example, if 1 m/yr thinning began in 1973, the change detected between 1973-1989 would be similar to the total change detected if 2 m/yr thinning began in 1981. Best we can say from this is that we know the thinning started sometime between 1973-1989, not that it started in the 1970s).

Referee #2 (Remarks to the Author):

Re-Review of Miles & Bingham, Nature, November 2023

Authors carefully examined all additional suggestions from the referees and this is now ready for publication. Only a few technical points below.

Line numbers refer to the track change version of the MS.

Everywhere, write rather "ICESat", i.e. using upper cases for "CE".

L121. "The only exception has been to Bawden Ice Rise, located". Is wording OK? "to"?

L231. "in" or "during" before "the late 1970s",

L270. authors used "mCDW" as acronym before.

Figure 1. I suggest having the panels reorganized to have a first row dedicated to ice shelf thickening (so actual panels a, d but the two sub-panels of d now organized horizontally) and a lower row with the panels illustrating thinning (b and c). Seems a more logical set up.

L392. The added sentence starting with "In a small..." is rather complex to follow. Split into two?

Ext Figure 4. Authors would need to tell somewhere (on the figure or in the caption) that the time series shown here is from radar altimetry which is different from ICESat laser altimetry. (Many reader may not be aware of this just from the fact that it comes from Ref 3).

Author Rebuttals to Second Revision:

Referee #1 (Remarks to the Author):

I think the discussion of the comparison against altimetry has improved. My points in the last review were meant to pick on the the equating of rigor of the methods (highly accurate, but not error-free laser altimetry vs non-optimized imagery, which understandably is tricky when you have one/two clear images a year and no control over sun-angle and other factors) rather than questioning why altimetric data might see things differently, but from the response to referees and the manuscript adjustments, my comments may have been interpreted to suggest that the altimetry results needed more explanation. That wasn't entirely my intent, but the authors now include some theoretical considerations that I think fit well here anyway, so the additions are well taken.

I think overall the manuscript has improved and provides useful results that are of general interest to the scientific community and specific interest to the glaciological community overall. However, I do have some concerns about the over-interpretability of limited observations, which is not a fault of the work specifically, it's just a fact that we don't have sufficient observations over long timescales. These authors have maximized the use of what does exist to interpret Antarctic ice shelf thinning over three ~decadal epochs, which extends our understanding of how this has changed over the two decades prior to altimetry data becoming available. This is valuable. However, I would caution against over-stating the results as being a record of change over 50 years, which is the message that comes across in this manuscript. What it does do is provide ~decadal snapshots of change, the latter two of which (1989-2000 and 2000-2022) corroborate existing observations of ice shelf thinning measured by altimeters (~1992-onwards). Perhaps a subtle point and I do not mean it to take away from the impact of this work, but I do think it's important to ensure that interpretations have some guardrails which are a bit buried in the text for now. This is particularly exemplified in the title, which suggests that ice shelves have been unanchoring from pinning points for the last 50 years, however a visual inspection of the resultant dataset in Figure 2 (and reading of the descriptive text) highlights that aside from the west Antarctic Peninsula, segments of the Amundsen Sea Sector, and Totten area, much of the ice shelf area around the continent either showed no decisive change or thickened in the first epoch (1973-1989). It is only in the second and third epochs (1989-2000 and 2000-2022) that unanchoring/thinning becomes more widespread, and only in the most recent epoch that I would judge unanchoring to be the predominant observation. If we had data to compare from the 1960s to show unanchoring increased in the 1970s, then we could state that the ice shelves have been unanchoring for 50 years. But that data does not exist, and as such, this methodology can only say for certain that unanchoring increased between the 1973-1989 period and the 1989-2000 period (and further into the 2000-2022 period) - that's more like a ~30 year inference rather than a 50 year inference (i.e., change over the 1973-1989 period do not necessarily mean that the changes occurred in 1973 - they could have occurred in 1988). The fact that ice shelves have been thinning since first measured in ~1992 has already been established with altimeter measurements. This is definitely not intended to say that altimeters are the be-all, end-all, but certainly this has been observed previously. In that way I think the title is exemplary of the way in which this study somewhat overstates its interpretability. I don't disagree that we should sound the alarm on these decadal scale changes and I think the effort on behalf of these authors is very worthwhile, but we are also in a time at which we don't need to be overstating impacts because that inevitably makes it harder to communicate with clarity.

We again thank the reviewer for taking the time to offer constructive feedback on our manuscript. We have lightly edited the manuscript including the Abstract/summary paragraph to make it clear that our work focuses on decadal snapshots.

Specific comments:

Editorial note, I was trained under the notion that abstracts should not have references within them as they are a summary of the work to be discussed. This abstract has references, I assume perhaps Nature has different formatting guidelines?

We believe that this is consistent with editorial guidelines.

Line 28: “attributed to warming ocean currents” - Perhaps a semantic point, but it’s not the ocean currents that are warming. It’s the subsurface bodies of water carried by the currents. I also might be relevant in this area to mention that the majority of ocean-warming-induced acceleration of ice discharge has been in West Antarctica (of course with a few examples in EAIS), to place this statement in context with observations given the interest in WAIS change, and to set up the background and contrast for what the new findings in this manuscript show in terms of Antarctic-wide change eg in Lines 100-102.

Amended. We have change to text to ‘warm ocean currents’ and highlighted that this is mainly happening in West Antarctica and the Wilkes Land coastline of East Antarctica.

Line 71: “ICESat-1” should be just “ICESat”

Amended

Line 190-192: I think the language specifically here and generally in the manuscript is a bit over-stated. The results described in this manuscript relate to changes between three epochs (1973-1989; 1989-2000; 2000-2022). Stating that the results show some ice shelves were “already thinning in the 1970s” is not entirely representative of what the methods employed here can show. Because of the snapshot-like approach dictated by the (understandably) limited data in the 70s and 80s (and 90s), we can say that thinning occurred sometime between 1973 and 1989. Technically, the majority of the intervening years are in the 1980s. I am not sure how it can be said with certainty that thinning began in the 1970s; I don’t disagree that it’s likely, but that’s not what this method demonstrates. In all, I don’t think this statement is wrong, but I do think it’s misleading, and further interpretation of that to mean that triggers existed prior to the 1970s might be a stretch. I think this needs to be couched a little bit more in terms of what interpretations the methods actually allow for vs assuming a change between 1973-1989 was due to changes in the 1970s. This goes more broadly for the manuscript as a whole, as well - while it does enable a ~50-year lookback that hasn’t previously been investigated, the methods presented enable an inter-epochal comparison over a 50-year period, rather than a 50-year time series of change, in essence adding two ~decadal-scale periods of information (1973-1989; 1989-2000) to the existing 30 year annual/sub-annual time series record. I think this point is somewhat muddled throughout, including in the title and abstract. I don’t mean for this to be a critique of the method or the results, as they are valuable; it’s just the language is a bit over-stated in terms of what we can actually interpret from decadal-scale change observations (for example, if 1 m/yr thinning began in 1973, the change detected between 1973-1989 would be similar to the total change detected if 2 m/yr thinning began in 1981. Best we can say from this is

that we know the thinning started sometime between 1973-1989, not that it started in the 1970s).

We agree with the reviewer, as such we have amended this particular sentence to:

“Our results show that at least parts of their ice shelves were already thinning between 1973 and 1989”

We have also made similar edits in other areas of the manuscript i.e. where we have referred to the 1970s instead of 1973-1989.

Referee #2 (Remarks to the Author):

Re-Review of Miles & Bingham, Nature, November 2023

Authors carefully examined all additional suggestions from the referees and this is now ready for publication. Only a few technical points below.

We again thank the reviewer for taking the time to offer constructive feedback on our manuscript.

Line numbers refer to the track change version of the MS.

Everywhere, write rather "ICESat", i.e. using upper cases for "CE".

Amended.

L121. "The only exception has been to Bawden Ice Rise, located". Is wording OK? "to"?

We have removed “to”.

L231. "in" or “during” before "the late 1970s",

“In” is correct because the relic ice rise advected with ice flow once it ungrounded. Thus from the 1989 image we can work out that it ungrounded in the late 1970s.

L270. authors used “mCDW” as acronym before.

Amended

Figure 1. I suggest having the panels reorganized to have a first row dedicated to ice shelf thickening (so actual panels a, d but the two sub-panels of d now organized horizontally) and a lower row with the panels illustrating thinning (b and c). Seems a more logical set up.

This is a good suggestion and we have done this.

L392. The added sentence starting with "In a small..." is rather complex to follow. Split into two?

We have simplified the sentence.

Ext Figure 4. Authors would need to tell somewhere (on the figure or in the caption) that the time

series shown here is from radar altimetry which is different from ICESat laser altimetry. (Many reader may not be aware of this just from the fact that it comes from Ref 3).

We have clarified in the caption the difference between laser and radar altimetry.